# Alzheimer’s Disease, and Breast and Prostate Cancer Research: Translational Failures and the Importance to Monitor Outputs and Impact of Funded Research

**DOI:** 10.3390/ani10071194

**Published:** 2020-07-14

**Authors:** Francesca Pistollato, Camilla Bernasconi, Janine McCarthy, Ivana Campia, Christian Desaintes, Clemens Wittwehr, Pierre Deceuninck, Maurice Whelan

**Affiliations:** 1European Commission, Joint Research Centre (JRC), 21027 Ispra, Italy; camilla.bernasconi@ec.europa.eu (C.B.); ivana.campia@ec.europa.eu (I.C.); clemens.wittwehr@ec.europa.eu (C.W.); pierre.deceuninck@ec.europa.eu (P.D.); maurice.whelan@ec.europa.eu (M.W.); 2Physicians Committee for Responsible Medicine (PCRM), Washington, DC 20016, USA; jmccarthy@pcrm.org; 3European Commission, Directorate General for Research and Innovation (RTD), 1000 Brussels, Belgium; christian.desaintes@ec.europa.eu

**Keywords:** biomedical research, Alzheimer’s disease, breast cancer, prostate cancer, funding, indicators, translational failure, animal models, cross-disciplinarity

## Abstract

**Simple Summary:**

Noncommunicable diseases, such as Alzheimer’s disease, and breast and prostate cancer, are becoming increasingly prevalent in Western countries. To better elucidate the onset and evolution of these pathologies and ultimately design new preventive and therapeutic strategies, research activities focused on these biomedical areas have been supported by the European Union for more than two decades. While research has globally contributed to increase our understanding of the pathological mechanisms underlying these diseases, the failure rate in drug development still remains very high. Nowadays, it is important to monitor the contribution to innovation and impact of funded research by means of defined indicators.

**Abstract:**

Dementia and cancer are becoming increasingly prevalent in Western countries. In the last two decades, research focused on Alzheimer’s disease (AD) and cancer, in particular, breast cancer (BC) and prostate cancer (PC), has been substantially funded both in Europe and worldwide. While scientific research outcomes have contributed to increase our understanding of the disease etiopathology, still the prevalence of these chronic degenerative conditions remains very high across the globe. By definition, no model is perfect. In particular, animal models of AD, BC, and PC have been and still are traditionally used in basic/fundamental, translational, and preclinical research to study human disease mechanisms, identify new therapeutic targets, and develop new drugs. However, animals do not adequately model some essential features of human disease; therefore, they are often unable to pave the way to the development of drugs effective in human patients. The rise of new technological tools and models in life science, and the increasing need for multidisciplinary approaches have encouraged many interdisciplinary research initiatives. With considerable funds being invested in biomedical research, it is becoming pivotal to define and apply indicators to monitor the contribution to innovation and impact of funded research. Here, we discuss some of the issues underlying translational failure in AD, BC, and PC research, and describe how indicators could be applied to retrospectively measure outputs and impact of funded biomedical research.

## 1. Introduction

Noncommunicable diseases (NCDs) are becoming increasingly prevalent in Western countries, and globally account for more than 80% of total premature deaths [1]. NCDs are generally the result of a combination of genetic, physiological, environmental, and lifestyle factors, with both young and middle-aged people affected. The economic consequences are also important, with lowering of earnings, employment rates, and productivity; enormous societal impacts; and health care costs. The negative effects on the gross domestic product (GDP) for EU economies are estimated to be in the order of 115 billion euros, corresponding to 0.8% of GDP annually [2].

Among NCDs, dementia, including Alzheimer’s disease (AD), affects nearly 50 million people worldwide [3]. According to the 2016 World Health Organization (WHO) statistics, AD and other dementias represent the fifth leading cause of death globally [4], and since 2015, AD has become the first leading cause of death in the UK [5]. In 2015, the number of people living with dementia in Europe was estimated to be 10.5 million (with AD accounting for 60% to 80% of dementia cases) [6].

Cancer is the second leading cause of death globally, with nearly one in six deaths caused by cancer [1]. According to the 2018 world cancer statistics [7], about 18 million new cases were diagnosed in 2018 with cancer (9.5 million men and 8.5 million women), with lung cancer, breast cancer (BC), colorectal cancer, and prostate cancer (PC) representing the four most common cancers globally (Table 1).

In Europe (EU-28), in 2016, 97,000 people died from BC, making up around 7% of all deaths from cancer, and among women, BC accounted for 15.6% of all cancer deaths [8]. With regard to PC, 76,900 men died from PC in the EU-28 in 2016, which corresponds to 5.6% of all deaths from cancer [8].

Animal models of AD, BC and PC have been and are still largely used in some of those projects aiming to recapitulate human disease features, understand the underpinning molecular and cellular mechanisms, often with the goal of identifying new druggable targets and ultimately developing new drugs. Transgenic (Tg) animals (mainly rodents) are generally purported to be reliable proxies of human diseases [9], and ‘humanized’ animals accounting for more than one engineered genetic modification are deemed to be more suitable to study complex multigene disorders [10]. However, as a matter of fact, animal models typically do not develop the diseases as they physiologically occur in humans, and the successful translation of novel research on preclinical animal models to the clinic often remains a significant barrier [11,12,13], as it has been discussed mainly in the context of AD [14,15,16], but also with regard to BC [17,18] and PC [19,20]. Flaws in animal experimentation design; inappropriate target selectivity; neglecting efficacy, pharmacodynamic, and pharmacokinetic properties of new compounds, along with inappropriate selection of clinical trial participants are among the possible reasons behind the clinical failure rate in drug development, which still remains very high, with an overall likelihood of approval from phase I of about 9.6% [21]. Notably, 97% of drugs that are tested in clinical trials in oncology never advance to receive regulatory approval [22], with a lack of efficacy and (off-target) toxicity representing the most common causes of trial failure. With regard to AD, the failure rate of drug development reaches 99% [23], and zero disease-modifying therapies have been developed so far [24]. One of the main reasons of these failures is linked to the current drug development pipeline, which still strongly relies on animal models at the preclinical stage [25].

In an effort to develop and optimize more human relevant models and increase predictive capacity, a wide range of non-animal approaches have been developed in recent years, spanning from patient-derived cells, such as induced pluripotent stem cells (iPSCs) [26,27,28], three-dimensional (3-D) tumor spheroids [29], complex microfluidics organ-on-chip technologies [25,30], to next-generation sequencing and omics technologies, integrated computer modelling, systems biology, and imaging techniques [31,32]. These models and techniques, along with data derived from clinical and observational studies, can already be used in an integrated manner to gather insights into disease molecular and cellular mechanisms, discover new biomarkers, and design novel therapeutic and preventive strategies.

The rise of new technological tools and models in life science, and the increasing need for multidisciplinary approaches, have encouraged many research initiatives and the launch of numerous research projects funded by the European Commission (EC), in particular under Framework Programme 7 (FP7: 2007–2013) and Horizon 2020 (H2020: 2014–2020), to further develop such innovative approaches [33,34,35]. During the same period, research on NCDs, and in particular on AD, BC, and PC, has been substantially funded by the EC (Figure 1, Figure 2 and Figure 3). This research effort has contributed to numerous flagships of health research, such as the Innovative Medicines Initiative [36], which has globally aided in increasing our understanding of the mechanisms underlying disease etiopathogenesis and consolidation, with a large portfolio of different activities, including basic understanding, diagnostics, drug development, drug repurposing, clinical trials, etc. Nowadays, it is becoming pivotal to define and apply indicators suitable to monitor the contribution to innovation and the impact of funded research. In the next sections, we highlight some of the issues hampering the translation of biomedical research results, specifically in the field of AD, BC, and PC. These diseases were selected as representative case studies for NCDs for several reasons: (i) Their high prevalence; (ii) the high number of animals used, as indicated by the most recent EU statistics on the number of animals used for scientific purposes [37]; (iii) the important research investments; and (iv) the high level of translational failure in drug development.

We also present some recent initiatives being undertaken by the European Commission’s Joint Research Centre (JRC) in collaboration with EC Directorate General for Research and Innovation (RTD), aimed at defining indicators to retrospectively monitor the outputs and impact of EU-funded biomedical research. Finally, we briefly discuss how human-specific (non-animal) methods could already be used to advance basic and translational/applied research in the field of AD, BC, and PC, and could be used to advance drug discovery and testing.

## 2. Three Biomedical Research Areas Characterized by a High Rate of Translational Failure: Alzheimer’s Disease, Breast Cancer, and Prostate Cancer

### 2.1. Alzheimer’s Disease

According to the WHO, the proportion of the world’s population over 60 years of age will nearly double from 12% to 22% between 2015 and 2050 [38]. Each year, 9.9 million new cases of dementia are registered worldwide, which corresponds to nearly one new case every 3.2 s [39]. Today, dementia affects nearly 50 million people globally, with significant public health costs [40]. Among dementias, AD accounts for about 60–80% of total cases [3].

The most common form of AD (95–99% of all AD cases) is the late-onset type (LOAD), often sporadic, generally occurring after 65 years of age, while about 1–5% of AD cases are early onset, including familial (genetically driven) (FAD) and sporadic forms. Early onset FAD generally appears before 60–65 years of age, and often before 55 years of age. Three genes are known to cause FAD: Amyloid precursor protein (APP), presenilin-1 (PS1), and presenilin-2 (PS2), with PS1 mutations accounting for most of early onset FAD [41]. Additionally, the ε4 allele of apolipoprotein E (APOE4) is a known genetic risk factor, which increases the likelihood to develop LOAD, with homozygous APOE4 carriers being nearly 15 times more likely to develop the disease [42].

All AD forms are generally characterized by the accumulation of amyloid beta (Aβ) plaques and hyperphosphorylation of tau, the major microtubule-associated protein, causing neurofibrillary tangle (NFT) formation. The disease develops with progressive cerebral atrophy, cognitive function decline, and ultimately death [43,44]. However, the role of some etiological mechanisms and contributing factors implicated in the onset and the consolidation of AD, both FAD and LOAD, still remain not fully elucidated.

Concerning drug development, despite very encouraging results obtained in preclinical animal models, showing a reduction of NFTs, APP, or Aβ plaques, often accompanied with significant improvement of spatial learning and memory, therapeutic approaches based on the ‘amyloid cascade’ hypothesis or designed to target tau proteins have generally failed to provide beneficial effects in AD patients [45,46]. One possible explanation for the failure of clinical trials is the time when drugs are being given, considering that Aβ plaques can form decades before the appearance of first cognitive symptoms [47]. Further complicating the relationship between Aβ plaques and AD, studies on post-mortem brains have reported zero or minimal levels of brain Aβ plaques in about 14–21% of clinically diagnosed patients [48,49,50].

To date, only five drugs, i.e., memantine (an antagonist of the N-Methyl-D-Aspartate-receptor subtype of glutamate receptor), three acetylcholinesterase (AChE) inhibitors (donepezil, galantamine, and rivastigmine), and a combination of donepezil with memantine, have been officially approved for the treatment of AD [15,51]. While these drugs can help temporarily improve cognitive and behavioral symptoms in AD patients, they have no effects on the long-term prognosis, they are not effective in all patients, and those patients who will respond to treatments cannot be identified early, as reported for rivastigmine, galantamine, and donepezil [52]. As indicated in a systematic review and meta-analysis study on the effects of donepezil in 8257 participants with mild, moderate, or severe AD, the quality of the evidence of donepezil’s beneficial effects on cognitive function and daily living activities, after 12 or 24 weeks of treatment, has been rated as moderate, generally due to study limitations [53].

In the last 15 years, there have been no approved disease-modifying treatments for AD. In a 2019 analysis of the drug development pipeline for AD (conducted by using the clinicaltrials.gov database), 132 agents were in clinical trials for the treatment of AD (28 agents were in 42 phase III trials). Of the several drugs that have completed clinical trial evaluation since the 2018 pipeline analysis conducted by the same group, none of them have shown a drug–placebo difference. Those clinical trials have been terminated, often upon futility analysis (which probes the ability of a clinical trial to achieve its objectives [54]), such as for crenezumab, aducanumab, verubecestat, lanabecestat, intranasal insulin, pioglitazone, AZD0530, and ITI-007; others have been terminated due to the appearance of adverse effects (e.g., atabecestat). Of the 17 disease-modifying treatments that were in phase III according to the authors’ previous 2018 review, 8 were terminated (as of 12 February 2019) [55].

The reasons underlying failure in AD drug development may be associated to some major issues, as highlighted by Mullane and Williams in their recent perspective article [15]. In particular, during preclinical drug assessment, some relevant aspects, such as target selectivity, efficacy, and the pharmacodynamic and pharmacokinetic properties of new compounds [56,57,58,59], are sometimes neglected or not taken into sufficient account, making compounds proceed to later phases of clinical trials even in the absence of clear evidence of drug efficacy [60,61]. For instance, this has happened for compounds targeting amyloid deposition, which, despite promising effects in preclinical animal models, have been shown to be ineffective in clinical trials [14,62].

Along with this, there has been the tendency to conceive AD exclusively as an amyloid- and NFT-related disease, and therefore to design drugs aimed at targeting these pathological features, not reckoning the multifactorial nature of this disease, encompassing metabolism, immune system, and inflammation, along with environmental and lifestyle-related factors [63].

Additionally, it is generally assumed that LOAD, the most prevalent AD form, and the less frequent early onset FAD are essentially the same type of disease. As both FAD and LOAD are characterized by the same pathological and behavioral traits [64], the development of drugs or new druggable targets suitable for LOAD have been studied in Tg mouse models that overexpress human genes associated with FAD. However, these models have been shown neither to recapitulate the genetics, nor the onset and progression of LOAD as observed in patients [14,65,66]. Tg and double-Tg animals develop Aβ plaques and associated brain inflammation, undergoing cognitive and behavioral deficits. These models do not produce NFTs, which can be observed along with cognitive deficits in animals engineered to express the mutated tau protein. The triple 3xTg mice (mean lifespan of 12–18 months) express mutated human APP, PSEN1, and tau protein and can generate both Aβ plaques and NFTs, producing also gliosis, synaptic deficits, and memory impairment [67].

Of the 180 available mouse models of AD (Appendix A) [68]), very few are representative of LOAD [69]. One of them could be the senescence-accelerated mouse prone 8 (SAMP8), with a mean lifespan of 9.7 months, and characterized by a spontaneous accelerated aging phenotype, which is considered more suitable to study brain ageing and LOAD [70]. SAMP8 mice develop Aβ plaques, NFTs, and hyperphosphorylated tau, exhibiting spongiosis, gliosis, forebrain cholinergic impairments, and dendritic spine loss [71]. Additionally, variant strains of the SAMP8 model (e.g., SAMP8-APP, SAMP8-PS1, and SAMP8-APP/PS1 models) [72], and the more recently developed App KO/APOE4/Trem2*R47 mouse model [73], represent additional alternate models considered suitable to study LOAD.

Currently available treatments have shown beneficial effects in SAMP8 mice. For instance, memantine was found to improve spatial learning and memory and reduce both hippocampal CA1 NFTs and APP levels when administered to SAMP8 mice; additive effects were observed by combining memantine with environmental enrichment [74]. Along the same line, donepezil was found to improve spatial learning and memory ability, increase cerebral glucose metabolism, and reduce Aβ levels in the cortex of SAMP8 mice. When combined with manual acupuncture, additive beneficial effects were observed [75]. However, despite the promising results obtained in preclinical trials, clinical trials have not proven significant beneficial effects, especially in the long term. In particular, memantine treatment has shown unclear positive effects in patients, slowing the process of cognitive loss at most [76]. Combination therapies are considered more promising than individual treatments in slowing cognitive decline; for example, administration of memantine, in combination with AChE inhibitors (e.g., donepezil or galantamine) was shown to provide some behavioral benefits in patients affected by moderate to severe AD [77,78].

Despite the extensive characterization of the SAMP8 and the SAMP8 murine variants, the genes responsible for the accelerated senescence and the exhibited pathological features are almost unknown. Moreover, Aβ plaque formation and cognitive abnormalities in these mice appear to be significantly different from human AD [79].

Additionally, Tg animals expressing FAD genetic variants (Appendix A), despite showing amyloids, NFTs, gliosis, and synaptic alterations, often do not undergo significant neuronal loss, and the amyloid peptides they generate appear to be different from those identified in the human brain [80]. Another relevant aspect is that animal models of AD generally do not reflect the pathology as observed in humans [16,81], and do not develop the typical co-morbidities observed in AD patients, such as metabolic syndrome, cardiovascular disease, inflammation, and immunological disorders [15]. Notably, different murine strains show remarkably different lifespans, often premature mortality, along with sex differences in the expected lifespan, as summarized by Rae and Brown [82]. Although efforts to translate lifespan developmental stages in mice to the equivalent stages for humans have been made considering chronological ages [83], these comparisons have been based on the C57 black 6 (C57BL/6) mouse, which is one of the murine strains with the longest lifespan [84]. These differences make direct comparison between preclinical studies using different murine strains quite hard, and the translation of mouse data to human clinical studies questionable. Altogether, these animal–human discrepancies have contributed to making basic science research outcomes poorly applicable to human AD [50].

### 2.2. Breast Cancer

Nowadays, the average 5-year survival rate for BC is 91%, compared to the 53% of those diagnosed in the 1970s [85,86]. Despite this positive trend, BC still remains the most commonly occurring cancer in women, with over two million new cases and more than half a million of diagnosed women succumbing every year [87].

To date, there are 34 drugs approved for BC treatment [88,89], a relatively high number, compared to other solid tumors. However, only 5% of molecules that show anticancer activity in preclinical studies are approved upon demonstration of sufficient efficacy in phase III clinical trials [90]. This trend is extremely prevalent especially for vascular endothelial growth factor (VEGF) inhibitors (e.g., Bevacizumab), which are also used for BC treatment [91]. As reported for AD [56,57,58,59,60,61], incorrect identification of drug targets, improper proposed mechanism of action, drug toxicity, drug resistance, and weak genetic evidence are among the possible failure reasons [22].

For example, although hormone therapies (e.g., tamoxifen, inhibitors of aromatase enzymes involved in estrogens synthesis) and Herceptin (trastuzumab) have improved clinical outcome of poor prognosis for estrogen receptor (ER)-positive and human epidermal growth factor receptor 2 (HER2)-positive cancers, respectively, many patients develop progressive disease [92,93,94]. This suggests that, despite drugs successfully passing preclinical and clinical phases, reaching marketing approval, they may still prove ineffective in the long term. Assessment of drug efficacy in the long term would not be possible in animal models, considering the way preclinical animal experimentation is generally designed (i.e., based on animals, such as mice, with limited lifespan, and treated for relatively short periods of time), which does not allow the prediction of possible drug resistance and disease progression.

Understanding de novo or acquired resistance to these drugs is one of the biggest challenges in the identification of new effective therapeutic agents. As commented by Moissenko et al. in their perspective article [95], both intracellular (e.g., drug metabolism and efflux, target modulations, lesion restoration) and extracellular mechanisms (e.g., crosstalk between tumor cells and environmental factors) may be responsible for drug resistance in BC. Although several mechanisms underlying tumor cell resistance to conventional cytotoxic compounds have been elucidated, more research is warranted to elucidate how multidrug resistance occurs in patients with advanced BC [95].

The story of iniparib (a poly (ADP-ribose) polymerase 1 (PARP1) inhibitor) teaches us the importance of comparators and specialized populations’ selection in clinical trials. Combined with chemotherapy, iniparib increased the rate of response to 52% (from 32% in the chemotherapy-alone group), suggesting that it may overcome the intrinsic drug resistance of some triple-negative BCs [96]. However, these promising data were not confirmed in the phase III trial. The fact that patients in the control group of this trial were permitted to crossover to iniparib has been implicated as potentially biasing the overall survival results. Further studies suggest that the actual mode of action of iniparib is different from what was originally expected and that its beneficial effects are largely restricted to *Breast Related Cancer Antigen (BRCA)*-mutation carriers [97]. Iniparib is a stark example of how incorrect interpretation of preclinical data in animals, along with poor clinical trial design, may skew results interpretation and decision-making, leading to late-stage trial failure. As highlighted by Mateo et al. [98], collected preclinical data could not elucidate the mechanism of action of iniparib before the initiation of clinical trials, and phase I trials did not prove the mechanism of action of this drug. Additionally, inappropriate selection of patients and the lack of implementation and validation of predictive biomarkers can further contribute to clinical failure. These are just some of the critical factors to be carefully considered during the development of anticancer drugs, in order to minimize failures in future late-stage clinical trials.

The greatest challenge to defeat BC is linked to the high tumor heterogeneity. Indeed, it is considered a combination of heterogeneous-related diseases, each with its specific histopathological, genomic and proteomic characteristics [99]. Heterogeneity seems to be more frequent within, not across, different BC subtypes: Only a small part of the mutations found in the primary tumor are detectable in the metastatic lesion, indicating significant genetic evolution occurring during the metastatic process [99].

Clinically, BC is categorized into three basic therapeutic groups: (i) The ER positive, (ii) the HER2 positive, and (iii) the triple negative (ER/PR/HER2 negative, where PR stands for progesterone receptor) for which no targeted therapy is currently available [100], besides chemotherapy.

Although advances in sequencing analysis have enabled the identification of many mutations, their role in disease progression is not fully understood. The Cancer Genome Atlas (TCGA) [101] has molecularly characterized over 20,000 primary cancers, and matched normal samples spanning 33 cancer types. These data, in combination with validated in vitro or in silico studies, could provide a powerful tool to explore important genomic trends or individual genes involved, possibly enabling personalized medicine approaches.

The key to success in drug development and clinical treatments is the availability of reliable preclinical models that accurately recapitulate the relevant clinical features of the disease, and therefore allow reliable screening of anticancer agents with robust clinical correlation. These models should be able to catch the whole complexity and heterogeneity of BC. Beside cell cultures, a large number of animal models are available, reflecting different types and stages of the disease. It is important to select the correct model depending on the research question(s): Characterization of the different stages of the disease, the role of the immune system and of the microenvironment, and the metastatic disease and the pathway(s) responsible for drug resistance. Advantages and pitfalls of commonly used BC models are commented in [102,103] and briefly described below.

BC cell lines have been largely used since the 1970s as in vitro models for drug discovery [104]. Cells are cultivated in an artificial environment that usually select specific populations and induce changes to facilitate cell adaption to the artificial culture environment (i.e., plastic). Continuous cell passaging might cause clonal selection and consequent loss of heterogeneity following disruption of the original tumor structure and microenvironment. Gisselsson et al. [105] comprehensively reported about the possible causes of clonal selection, specifically identifying (i) alterations in telomere function occurring over prolonged in vitro culture, and (ii) population (or genetic) bottlenecks (i.e., a significant decrease in the size of a biologically reproductive population that could be caused by factor(s) limiting the number of cells allowed to proliferate) as frequently neglected phenomena that may cause alterations in (cancer) cell line genotypes even after few passages in vitro. Compared to patient tumors, BC cell lines show a higher mutation frequency, which, over many in vitro passages, may originate a cell different from the original one. Mutations and chromosomal instability can trigger genomic heterogeneity, altering the transcriptional profile and drug response of cell lines. As a consequence, a candidate drug might be effective on the selected cellular population but fail in the clinical trial. This could explain the contradictory results observed by comparing data obtained using the same cell lines in different non-validated studies [106]. Additionally, cancer cells cultivated in vitro do not include key factors, such as stromal cells, immune and inflammatory cell infiltration, and vascularity, whose finely regulated interplay is responsible for tumor growth and metastasis formation.

Cell line-derived xenograft (CDX) models are generated by transplanting immortalized human cancer cell lines into immunocompromised mice. The choice of the transplantation site, i.e., ectopic (via subcutaneous injection) or orthotopic (in the mouse mammary gland), is critical, as each site has its own microenvironment and vasculature affecting the tumor growth rate and drug delivery [107,108]. However, CDX models lack the broad molecular transformation events (intratumoral heterogeneity) that occur in human tumors and the organotypic tumor microenvironment; therefore, they cannot recapitulate what is observed in patients with particular respect to drug response, and have been shown to poorly predict clinically effective therapies [99]. CDX models very rarely develop spontaneous metastases, making their use to study BC metastasis questionable [102]. Moreover, the cell lines used to generate CDX are generally obtained from highly aggressive tumors or fluids that have been drained from lung metastasis (e.g., MDA-MB-231 cells), which make these models less suitable to studying early events in the evolution of the primary tumor [102]. Furthermore, CDX models seem to be more responsive to antiproliferative agents than primary tumor [107].

Despite these limitations, CDX models are still widely applied at early stages of in vivo studies both in academia and industry for their user-friendly technique and high reproducibility, and several CDX mouse models have been made available by animal models’ providers (e.g., [109,110]).

Patient derived xenograft models (PDX) are obtained by surgical implantation of patient-derived tumor explant into an immunocompromised mouse. Although this method does not require preculture of patient cells, it requires fresh patient material and operator expertise, is invasive, and is rather expensive [99]. Months are needed to establish the engraftment and develop preclinical research samples, a period often too long for clinician decision-making (weeks). Compared to CDX models, PDX models retain the genetic intra/intertumor heterogeneity of the original tumor. They reflect more accurately the human situation due to contextual incorporation of human stroma and associated vasculature and tumor-associated immune cells; although, after three to five passages following engraftment, the replacement with murine stroma represents an issue [111,112]. PDX models seem to recapitulate human tumor angiogenesis [113]; therefore, they are often used for evaluating antiangiogenic therapies. The predictive power of PDX models has led to co-clinical trials, where patients and PDX models implanted with the patient tumor are treated simultaneously [113]. Immunocompromised hosts, such as severely compromised immune-deficient (SCID) mice, non-obese diabetic (NOD)–SCID mice, athymic nude mice, recombination-activating gene 2 (Rag2)-knockout mice, and the NOD/SCID/IL2Rγc−/− mice, are frequently used to generate the PDX model of BC [114], as they allow tumor engraftment. However, as these models lack immune system cells, they are not suitable for the preclinical testing of immunotherapies [99]. Murine strains with a human immune system (“humanized mice”), generated by engrafting different types of human leukocytes and purified human hematopoietic stem cells (CD34+) obtained from bone marrow, umbilical cord blood cells, fetal livers, or thymus tissues, are currently regarded as suitable models for immunotherapy efficacy testing [114]. However, co-engraftment with human immune cells still presents some limitations (e.g., xenogeneic graft-versus-host responses) [115], and introduces an extra layer of complexity and costs associated with their generation and maintenance [116,117]. In PDX models, aggressive tumor subtypes are usually favored and these forms might not respond to therapy as the less aggressive forms. As they are generated at the point of BC surgery, only after a period of at least five years it would be possible to compare the PDX model’s capacity to form metastases to what is observed in the patient. The average mouse lifespan is about 2 years, and, in the case of the commonly used NOD SCID murine strain, the lifespan is approximately 30 weeks, which is often due to the spontaneous development of thymic lymphomas. Therefore, these models are not deemed suitable for long-term xenotransplantation studies [118].

At a clinical level, metastatic lesions are often vastly different from their primary tumor counterparts at both molecular and histological levels. Lungs and lymph nodes metastases are commonly observed in PDX models, while the high frequency of brain and bone metastases observed in patients (approximately 70%) are rarely observed [102,119].

Genetically engineered mouse models (GEMMs) are generally used to address early events in the tumor process, as these models spontaneously exhibit tumor initiation driven by oncogenes within the correct microenvironment (e.g., MMTV-PyMT). However, the lineage/expression domains of the regulatory sequences used to induce transgene expression are not well defined, and GEMM oncogenes may not necessarily be representatives of those observed in human tumors [102]. Specific Tg animals have been developed in an effort to emulate more closely the genetics of human BC, accounting for the temporal and spatial activation of specific oncogenes and the deletion of tumor suppressors targeted to the murine mammary gland, such as the Cre/*loxP* conditional *BLG-Cre;Brca1^F22−24^;p53KO* model [120]. Mouse and rat models of BC have been customized using the nuclease-based system Clustered Regularly Interspaced Short Palindromic Repeats (CRISPR)/CRISPR associated protein 9 (CRISPR/Cas9), which can target any gene within a eukaryotic genome, and have been made available by different animal vendors. Comprehensive lists of BC GEMMs along with their characteristics have been described in several review articles [121,122,123,124,125].

Considering the very high level of tumor heterogeneity observed in BC, with specific histopathological, genomic, and proteomic features, and the genetic evolution frequently observed during metastasis [99], the use of animal models (in particular, CDX and GEMM models) to test new drugs may not be the best methodological approach to account for this complexity and understand BC biology and evolution as it occurs in humans.

None of the above-mentioned models seem to successfully recapitulate the metastasis process in BC. Although the 5-year survival rate for metastatic BC is 22% [86], and there is reasonable evidence that some patients with metastatic BC can be cured [126], less than 5% of global funding for BC research is dedicated toward understanding metastatic BC, or finding solutions to extend the lives of metastatic BC patients [127].

By the time a woman is diagnosed with metastatic disease, her original biopsy tissue may no longer be available. In addition, it is often hard to get a sample of a metastatic tumor, which may be buried inside the brain or in an anatomical area that is difficult to access safely. Current metastatic therapies are expensive, often toxic, and subject to the eventual development of drug resistance. Besides drug treatment, earlier detection of metastatic disease is of paramount importance in the prognosis of the patient. Several EU projects have been funded for the development of advanced technologies, such as positron emission tomography/computed tomography scans, or to improve imaging resolution [128]. Besides metastasis detection, these imaging technologies could also help in the early detection of tumor, saving many lives [129]. The possibility of measuring circulating tumor DNA, although not very informative in terms of time for metastatic disease development compared to cancer antigen 27-29 (CA 27-29) detection, might allow effective therapy development for a (micro/early) metastatic lesion [130].

### 2.3. Prostate Cancer

Prostate cancer is the second most frequent diagnosed cancer in men and the fourth most commonly occurring cancer overall (Table 1). Annually, there are more than 1.2 million new cases of PC and over 350,000 deaths, with higher incidence rates and prevalence in developed countries [131]. PC incidence and mortality rates are strongly related to age, with the highest incidence seen in elderly men (>65 years of age). The etiology of PC remains largely unknown compared to other common cancers; however, risk factors include advanced age, ethnicity, genetic factors, and family history [132].

Differences in the incidence rates worldwide are likely due to the use of different diagnostic testing and advanced screening in different regions of the world. Despite advances made in the field, PC is still a major cause of morbidity and mortality, and the overall high global incidence rate stresses the need to strengthen prevention measures, diagnostic tools, and novel therapies to reduce the public health impact of this disease.

The field of PC research continues to be hindered by the lack of human-relevant preclinical models to study disease development and progression and to further knowledge of effective prevention and therapeutic strategies [133]. PC is a highly heterogeneous and complex disease, and the research has been obstructed by the use of imperfect animal models that do not accurately recapitulate the illness, are costly, take considerable time to develop, and fail to mimic the multifaceted aspects of the condition. The most widely used animals in human PC research include genetically engineered mice, xenograft mice, Tg rats, and dogs [134]. Nevertheless, all these models have limitations that present serious challenges to preclinical drug development and biomedical research. Differences in size and physiology, as well as variations in the homology of targets between animals and humans, have led to translational limitations. For instance, mice used in PC research rarely develop tumors that metastasize, making it almost impossible to study the terminal lethal events in cancer progression [135]. Other than humans, dogs are the only other species that develop benign prostatic hyperplasia and sporadic PC; however, male dogs experience a very low incidence of spontaneous PC and are generally androgen independent and lack a functional androgen receptor (AR), unlike PC in humans [134].

When taking a closer look at the most extensively used animal in PC research, the mouse, there are several disadvantages that may explain why so few drugs that work in mice do not work in humans. With regard to prostate gene expression, while both mouse and human prostate cells respond to androgen stimulation and signaling [136], the expression of certain human androgen-responsive genes (e.g., prostate-specific antigen (PSA) and prostate-specific membrane antigen (PSMA)) is not present in mice [137,138,139]. When analyzing the prostates of mice and men, they significantly differ anatomically. Human prostates consist of a single lobe and three zones (central, peripheral, and transitional) surrounding the urethra at the base of the bladder. In mice, the prostate is comprised of four lobes located circumferentially around the urethra. Testosterone levels also fluctuate between the two species, and significant interspecies differences have been described with regard to the serum protein-binding affinity for androgens, regulation and function of hepatic steroid metabolizing enzymes, and testosterone biosynthesis and metabolism [140]. In mice, the total and free plasma testosterone levels may vary drastically between individual mice and between genetic backgrounds [138]. Additionally, the histopathology and timeframe of PC development is different in mice [141].

Despite the worldwide research endeavors, few findings have influenced the clinical management of the disease [19]. When it comes to drug discovery and development, CDX and PDX mouse models, where human PC cells are transplanted into immune-deficient mice, are the most often used. However, this is problematic for many reasons; as it has been already commented in Section 2.2 (on BC), research has shown that the immune system plays an important role in cancer progression and eradication, yet the immune system cannot be studied in immune-deficient mice. Another issue with using CDX mice is that they fail to reproduce the diverse heterogeneity observed in humans, partially due to the increased homogeneity of established cell lines after long-term in vitro culturing and the relative lack of cell lines in PC [138]. The use of immune-deficient xenotransplanted mice for the development of hormone ablation therapies presents significant limitations, as these animals present inadequate (low) levels of testosterone, not reflecting those found in normally ageing men [142].

Several GEMMs of PC have been developed, by modulating the expression of specific oncogenes or tumor suppressors, growth factors and their receptors, steroid hormone receptors, or regulators of cell cycle and apoptosis, as summarized in [143,144]. An example is provided by the PB-Cre4xPTEN^loxp/loxp^ GEMM model of PC, which is considered suitable to study prostate adenocarcinoma development, tumor progression, and metastasis [144]. Similarly, the Tg adenocarcinoma of the mouse prostate (TRAMP) model has been shown to recapitulate both the preneoplastic and metastatic stages of PC [145]. However, these models also present intrinsic limitations [146]. In particular, the TRAMP model develops primarily neuroendocrine tumors, is based on an androgen-dependent promoter, rarely undergoes bone metastasis, and exhibits relatively short kinetics opposite to the typically slow development of PC in humans [147]. Along the same line, the PTEN conditional model described above has been shown to develop senescence, which limits cancer progression, and does not develop metastases [147]. These differences in how PC spontaneously develops and evolves in humans and how it is artificially recreated in animals can possibly explain why drugs that are effective in mice (and other animals) are very often not successful in humans. It comes as no surprise that of the hundreds of drugs tested in mice to reduce tumor volume, only a few have translated into a drug effective for treating PC in humans [138]. As commented in Section 2.2., only about 5% of anticancer therapies tested in animals demonstrate sufficient efficacy in phase III clinical trials and are ultimately approved for clinical use [90]. The cost of these failures is estimated in the range of hundreds of millions of dollars per drug [148].

One drug, Orteronel (also known as TAK-700), is a hormonal therapy that was tested for the treatment of PC. Orteronel was designed to inhibit the 17,20 lyase activity of the enzyme CYP17A1, which is important for androgen synthesis in the testes, adrenal glands, and PC cells. Preclinical studies in animals provided the rationale for testing Orteronel in PC patients. In particular, Orteronel treatment caused a significant suppression of serum testosterone levels, shrinking several androgen-dependent organs in uncastrated rats [149]. Moreover, administration of Orteronel (twice daily) in intact cynomolgus monkeys induced a reduction of serum dehydroepiandrosterone sulfate and testosterone levels vs. vehicle control, and in castrated monkeys, such effects were even greater and persisted throughout the treatment period [150,151]. Despite these promising effects in animals, during phase II clinical trials, 34% of participants experienced serious adverse effects, and 96% experienced other adverse effects as a result of the treatment [152]. In a larger phase III study with 732 participants receiving Orteronel intervention, 52% were affected by serious adverse events and 96% experienced other adverse effects [153]. Ultimately, the drug did not move forward since it did not extend the overall survival of the patients and caused significant adverse effects.

Another drug that initially showed promise in preclinical studies but failed during clinical trials is AZD3514. AZD3514 is an oral drug that targets AR function, with a novel mechanism of action that can result in downregulation of AR protein. In several in vivo studies using rats, this drug significantly inhibited testosterone-induced growth of the prostate and seminal vesicles and also reduced tumor AR [154]. This drug went on to two phase I clinical trials, where only 13% of patients treated with AZD3514 had a greater than 50% decline in PSA and 17% of patients had decreased clinical indicators of soft tissue disease [155]. Despite several PSA responses observed using this therapy on animals, only marginal outcomes were detected at the clinical level and the development of AZD3514 has been discontinued due to toxicity concerns and significant adverse effects [156].

There are currently 15 approved drugs on the market for various treatment options, yet PC was still the cause of 358,989 deaths in 2018 [131,157]. Improving PC research to the point at which it translates to reproducible and effective success in the clinic is a particularly difficult challenge. Considering the heterogeneity and complexity of the disease, more resources should focus on the use of human biology-based research models complementary (or alternative) to animals, to develop promising therapeutics for all forms of PC.

## 3. The Need for Indicators to Monitor Innovation and Impact of Funded Biomedical Research

In the last two decades, the EU under FP5, FP6, FP7, and the more recent H2020 has invested significantly in research on AD, BC, and PC. The majority of these research activities focused on the understanding of the pathophysiology of the diseases, diagnostics, and preclinical studies. Animals were used, usually along with other non-animal approaches, in 19–64% of projects.

Concerning dementia, 614 projects on AD were funded with 947 million euros (Figure 1). Support has been provided to activities of basic/fundamental nature, as well as translational and applied, and clinical research. Animal models of AD have been used in several of those projects (between 41% and 51% of these projects, depending of the FP), generally in combination with other non-animal approaches (in vitro, in silico, clinical data) to explore the molecular and cellular mechanisms underlying AD, identify new druggable targets, and test new compounds.

In relation to breast cancer, the EU has funded 411 projects equating to 700 million euros (Figure 2). Between 19% and 44% of those projects also accounted for non-human animal models, generally in combination with other non-animal approaches (in vitro, in silico, clinical data).

For prostate cancer, the EU has supported research projects with an EU contribution of 288 million euros, with a relatively high percentage of those projects (between 37% and 64%) accounting also for the use of animals, generally in conjunction with in vitro, in silico, or clinical data (Figure 3).

Scientific outputs of these research endeavors are helping elucidate disease mechanisms, enabling the identification of new druggable targets, and the design of new prevention and treatment strategies.

Several genetic and also environmental and lifestyle risk factors have been shown to contribute to AD [158] and increase cancer risk [159,160,161], highlighting the role of prevention strategies to reduce disease risk. In light of this complexity, the use of reductionist research approaches aimed at dissecting the individual contribution of single genes or molecules to the onset of such complex multifactorial pathologies may be misleading, possibly contributing to the high translational failure rate observed in these areas of research [25].

In recent years, the advancement of big data, machine learning, and artificial intelligence, are progressively advancing the biomarker discovery paradigm toward systems-level screening and testing approaches, challenging small data research approaches traditionally used to target specific queries. The rise of new technological tools and models in life science, along with the increasing need for multidisciplinary approaches, have encouraged many research initiatives and the launch of several EU calls for proposals [33,34,35], which have stimulated the submission of thousands of research projects. It is important to define and systematically apply indicators to monitor the contribution to innovation and impact of funded research. Monitoring activities have been initiated already under the FP6, FP7, and H2020 funding scheme, with the goal of capturing the results of research activities, and assessing their impact on the European economy and society [162,163].

Recently, EC JRC, in collaboration with the EC RTD, has initiated an activity to define in more detail some possible indicators to monitor the contribution to innovation and the impact of biomedical research, understand what scientific methods and research approaches underpinned the advances made, and, in this context, assess the contribution of animal models. This initial proposal may help illustrate what type of indicators could be considered to retrospectively assess the impact of EU-funded research activities. Such indicators could be divided into the following major categories: Funding/economic, dissemination, scientific and technological, regulatory and policy, public and social engagement, and education, training, and job opportunities (Table 2). Strengths and weaknesses of the proposed indicators will be investigated as recommended by internationally recognized best practices [164]. Overall, different types of indicators used in a complementary way may help provide a more accurate monitoring of EC-funded biomedical research.

With regard to bibliometric indicators (dissemination category), special attention should be paid to the possible caveats at stake, especially considering the possible limitations underlying the citation indicator (indicator 6 in Table 2) and the possible issues with its interpretation and validity [165]. As highlighted by Aksnes and colleagues [166], several factors may undermine the use of citations as a measurement of performance, which are generally related to the citation process, such as (i) extensive self-citation rates, (ii) so-called ‘negative’ citations (to criticize, correct, and disclaim other works), and (iii) ‘citation circles’ (i.e., researchers citing one another’s work). While these issues are fundamentally inherent in the use of citations as an indicator, they can also be limited, e.g., by adjusting for self-citations, and crosschecking ‘negative’ citations.

Moreover, to gather feedback from EU research funding recipients, the JRC has recently conducted a survey [167] addressed to current and former participants of EC-funded research projects (under FP5, FP6, FP7, and H2020) in the fields of AD and other dementias, BC, and PC. The survey aimed at gaining insights and understanding related to: (i) How EU-funded projects have contributed to innovation and major scientific breakthroughs; (ii) how scientific results have been translated into socioeconomic impacts of benefit to the society; (iii) what ingredients determined the success of research projects; and (iv) what scientific methods and research approaches underpinned the advances made.

In total, 202 researchers participated in the survey, mostly from academia, who worked on basic, translational, and clinical research. Amongst the major outcomes of their research activities, the contribution to the development of new methodological approaches was indicated by the large majority of respondents. Almost all participants (93%) indicated either that their research had an impact beyond their project, or that an impact may be seen in the future. In particular, 46% of respondents claimed that their research had a positive impact on diagnostic or prognostic tools but also on treatment or prevention actions, the design of clinical trials, or new patents. Among the major drivers of research success, effective collaboration with project partners, multidisciplinarity, the design of the research strategy, and the international dimension of their project were considered. A factual summary report was published to summarize the main results of this online survey [168].

It should be considered that (research) surveys present some caveats; in particular, provided information is self-reported and, as respondents are aware of being the subject of the study, their replies may often be different than they might be in other circumstances. As a consequence, the research impact might be overrated [169].

To gain a more in-depth understanding on some of the aspects investigated in the survey, in particular those concerning the translatability of research, social impact, and lay public engagement, in-depth interviews with a number of survey respondents are currently being conducted by the JRC. This follow-up analysis will help clarify some of the replies obtained in the survey, dig more deeply into scientists’ opinions and perception of translational failure in biomedical research, and elucidate whether claims of innovation and impact success have been based on a particular method (animal or non-animal) simply because of its use or involvement in research activities. A synopsis report providing a more detailed analysis of the responses to both the survey and interviews will be published at the end of the process.

The results of the survey have provided valuable input into several activities the JRC is currently pursuing in collaboration with the DG RTD. Besides the development of the aforementioned indicators and the design of a methodology to assess the output and impact of EU-funded research in important disease areas, like AD and cancer, a recent JRC activity described in the report ‘Bridging Across Methods in the Biosciences—BeAMS’ [170] fosters crossdisciplinarity in biosciences to better inform policy and address societal needs. In this context, methods can be seen as a means to bridge across different scientific communities and facilitate more integrated approaches to problem solving and effective research strategies. Along this line, crossdisciplinarity should also be considered as a way to bridge the gap between scientific research and clinical practice, for example, by making innovation and scientific outcomes more tailored to patients’ real needs.

Future JRC activities will also aim to assess how EU-funded research contributed to innovation compared to research funded outside the EU by means of selected indicators (Table 2).

## 4. Discussion

Non-communicable diseases, such as AD and other dementias and cancer, are becoming increasingly prevalent worldwide, and are among the top five leading causes of death in EU and non-EU countries [1,4,5].

In light of their high prevalence, research on AD and cancer, specifically BC and PC, has been supported by the EC over past the 20 years, in particular under the two most recent FP7 and H2020, as reported in Figure 1, Figure 2 and Figure 3. However, it should be considered that to date, the H2020 framework programme is still ongoing, and therefore data reported in Figure 1, Figure 2 and Figure 3 are not exhaustive, being representative only of the number of projects (and funding) available at the time this analysis was conducted. This explains the only apparent decrease of funded projects under H2020. For this reason, possible speculations about decreasing or increasing trends in the percentage of projects accounting also for animal studies may be inaccurate or even misleading.

While EU projects usually do not go beyond phase II, they have significantly contributed to increasing our understanding of the pathophysiology of such complex disorders, in particular brain diseases, enabling the identification of biomarkers and pathways for effective and non-toxic therapies. Considering the involvement of environmental and lifestyle-related factors in the onset of these diseases, primary prevention strategies should be encouraged and put in place to decrease the prevalence of BC and PC [171,172], and also reduce the risk factors and comorbidities associated with AD and dementia [173].

Despite our increased understanding of disease etiology, the clinical failure rate in drug development both for cancer and AD remains dismal (overall 97% for cancer [22], 99% for AD [23,24]). Another element slowing the drug discovery process is that negative data derived from preclinical and clinical studies are often not published for years or not published at all. The highest attrition in drug development is observed in the first clinical proof-of-concept (PoC) study, where lack of efficacy or toxicity are frequently observed [174]. For this reason, making first clinical PoC validation studies publicly available might be key to reduce attrition. Several initiatives have been taken by large companies in the last decade to promote data sharing and transparency. Among these, the ‘Medical Publishing Insights and Practices (MPIP) Initiative’ [175], a collaborative effort among members of several pharmaceutical industries and the International Society for Medical Publication Professionals (ISMPP) [176] aims at promoting and increasing integrity, trust, and transparency in medical publications and general communication, and expanding access to research results. Along the same line, some peer-reviewed journals, such as the *Journal of Pharmaceutical Negative Results* [177] and the *Journal of Negative Results in Biomedicine* [178], focus on publishing original and novel research articles resulting in negative/null results. Other journals, such as *BMJ* [179] or *Plos One* [180], also accept the publishing of negative studies. A transparent sharing of data would prevent other scientists from wasting resources and time on similar studies, avoiding delays, and honoring the pact researchers have made with their study participants [181,182].

Nowadays, a wide range of different models, methods, and approaches can be used to tackle important research questions. As a general consideration, the type of scientific method used, for example, based on a human cohort study, an animal model, a cell-based assay, or an in silico/computational method, can influence the way in which research problems are both formulated and addressed. The current drug development pipeline still strongly relies on animal models at the preclinical stage and is based on experimental hypotheses formulated during basic/fundamental and translational research generally based on the use of in vivo and in vitro approaches. Among the possible reasons underlying translational failure in AD and cancer drug development could be the over-reliance on non-relevant-to-human models used to dissect individual contributions of single gene(s) or protein(s) to the onset of complex multifactorial human pathologies. In this context, over-reliance on models that lack predictive validity could be considered as one of the reasons behind the poor translation of significant research to the clinic [11,12,13,183]. In parallel, the target selectivity, efficacy, and pharmacodynamic and pharmacokinetic properties of new compounds are sometimes neglected or not taken into sufficient account.

Our analysis of the proportions of EU-funded projects in the field on AD, BC, and PC accounting also (but not exclusively) for non-human animal models (Figure 1, Figure 2 and Figure 3), does not allow conclusions to be drawn about possible links between methodological approaches selected in EU-funded projects and the translational failure characterizing these three areas of biomedical research. A retrospective assessment of the level of societal impact and innovation derived from those EU-funded projects (accounting and/or not accounting for animal-based studies) by means of indicators, as defined in Table 2, could enable possible correlations about adopted methodological approaches in research and translational outputs. As pointed out in Section 3, in-depth interviews with former EU-funded project participants are currently being carried out by the JRC to also define these possible associations.

While these indicators may not be suitable to gather insights about economic returns from funded research, as explored in other studies (e.g., [184,185]), they may help assess the contribution to innovation and translational impact of funded research, serving as an evidence base to inform research funding bodies in their decision making. Similar monitoring activities have been carried out using different strategies. For example, with specific regard to citation (biblio)metrics, Hutchins et al. built a machine learning system suitable for use to detect whether a paper is likely to be cited by a future clinical trial or guideline, in an effort to predict translational research progress [186].

The multinational study ‘Project Retrosight’ assessed the translational impact of basic and clinical research in the field of cardiovascular disease and stroke (considering 29 research grants awarded in Australia, Canada, and the UK) [187]. This project considered several parameters, such as the publication of papers, PhD supervisions, and development of scientific methods subsequently used by other researchers, as examples of academic impacts, and citation in policy documents or guidelines, intellectual property licensing, and changes in policy or practice as examples of wider impacts. Notably, Project Retrosight identified several factors associated with high and wider impact: (i) A clear clinical motivation in study design; (ii) the co-location of basic biomedical research in a clinical setting; (iii) strategic thinking by clinical researchers about translation into clinical practice; (iv) research collaboration between basic biomedical and clinical research; (v) the creation of international consortia; (vi) the engagement with practitioners and patients; and (vii) collaboration with industry. These findings may support research funders and policy makers in the design of future calls for proposals and the prioritization of research programs.

Another relevant aspect to consider when assessing the contribution to innovation and the impact of biomedical research is understanding what methodological approaches underpinned the advances made and, in this context, the specific contribution of animal-based approaches. Apart from scientific concerns, the use of animals in biomedical research has also raised ethical concerns [188,189,190], often driven upon ‘harm–benefit’ analysis, i.e., expected scientific benefits and societal impact should outweigh the expected harm to the animals [191]. The harm–benefit consideration is clearly a fundamental aspect to account for whenever planning and conducting animal experimentation. Along this line, prioritizing the principle of the 3Rs (replacement, reduction, and refinement) and, in particular, updating its strategic application in favor of replacement strategies relevant to human biology should be encouraged to advance biomedical research, as recently commented [192]. Notwithstanding, a survey aimed at assessing general attitudes to the 3Rs and animal use in biomedical research suggest a tendency by scientists to prioritize refinement over reduction, and reduction over replacement, an ‘upturned hierarchy’ of the 3Rs order as originally proposed by Russell and Burch [193].

With regards to the number of animals used in life science, in February 2020, reliable estimates of animal use in the EU during the years 2015 to 2017 have been made available by the EC, as mandated by the Directive 2010/63/EU on the protection of animals used for scientific purposes [194]. In recent years, this mandate has enforced a more comprehensive reporting framework from EU Member States. According to this recent report [37], the basic research areas that accounted for the highest numbers of animal uses were nervous system (22% of basic research uses), immune system (17%), and oncology (14%); the applied/translational research areas accounting for the highest numbers of animal uses were human cancer (27% of applied/translational research uses) and human nervous and mental disorders (14%). In basic research, severe procedures represented 9% of uses for both the nervous system and oncology, while in applied/translational research, severe procedures represented 9% of uses for human nervous and mental disorders and 7% for human cancers. Tg animals represented 45% of uses in basic research (63% for oncology, 52% for nervous system), and 25% of uses in translational/applied research (42% for human cancers and 29% for human nervous system and mental disorders). These numbers provide cumulative evidence that these areas of biomedical research (both basic and applied/translational), i.e., central nervous system disorders (including dementia and AD) and oncology (including BC and PC), globally account for a very significant proportion of animals uses and therefore would deserve attention and close monitoring.

The rise of new technological tools and models in life science has fostered the launch of several EU calls for proposals [33,34,35] and the submission of research projects characterized by interdisciplinary strategies. Over the last two decades, the EC has funded more than 200 projects based on the use of alternative approaches, including in vitro models (e.g., 3-D cell culture systems, engineered tissues, organ-on-a-chip models, body-on-a-chip, etc.), with a total funding of over 700 million EUR. The average annual budget for this activity gradually increased across the consecutive FPs, from 11 million EUR in FP5, to 32 million EUR in FP6, to 47 million EUR in FP7, and 50 million in H2020 (data retrieved from the CORDA EC database). While these numbers refer to the overall investments in the field of new approach methods (NAMs) across different areas of life science research (not exclusively AD, BC, and PC), they are indicative of a progressive increase in the use of NAMs and alternatives to animal approaches. In particular, patient-derived cells, such as iPSCs, are often used to study human diseases in vitro (disease-in-a-dish models), providing insights into disease mechanisms. Such human cell-based models can be used along with human biological samples (e.g., blood, serum, tissue biopsies) for the definition of early biomarkers of human diseases and the design of new therapeutics [26]. AD patient-derived iPSC models harbor translational potential [27], and are already permitting the identification of AD biomarker candidates [28]. Three-dimensional (3-D) tumor spheroids have been shown to more closely resemble the complexity of tumor tissue architecture and biology, and can be used to perform drug screening in vitro [29]. For instance, such models have been used to study drug resistance [195], drug response, and diffusion [196] in BC. Along the same line, 3-D models of PC, integrated in high-throughput testing platforms [197,198], are speeding up the drug screening process, and may prove useful tools for personalized medicine, while reducing costs and time [199]. Moreover, complex microfluidic organ-on-a-chip technologies have the potential to significantly improve and impact the drug screening process, enabling the recreation of the functionality of human organs in vitro [25,30].

Furthermore, next-generation sequencing, omics, integrated computer modelling, systems biology, and imaging technologies are already used to study pathways of diseases, identify new biomarkers, and evaluate the molecular (genetic and epigenetic) effects of new treatments, as reported for cancer [31,200] and AD [32,201,202].

After decades of biomedical research based on the use of different research methods, both animal and non-animal, the use of indicators, such as those defined in Table 2, may help retrospectively monitor the contribution to innovation and societal impact of funded research. Monitoring of research output and impact will also help build new proposals for the upcoming Horizon Europe funding program, and determine its impact assessment [203]. In particular, the metrics here proposed may be taken into consideration for the design of the strategic planning of Horizon Europe, contributing to the creation of future calls for proposals, helping prioritize emerging research topics, and also feeding into the mid-term review of framework programs.

## 5. Conclusions

Alzheimer’s disease, and breast and prostate cancer are among the top five leading causes of death worldwide, often occurring between 30 and 70 years of age, at a time in life when people are working and more productive for the whole society. Understanding the genetic and epigenetic insights and the molecular mechanisms of these diseases, translating this knowledge into clinical practice, and designing tailored screening approaches along with preventive strategies are probably the main challenges in biomedical research. The overall poor clinical translation of preclinical data traditionally generated by using animals along with oversimplistic in vitro models suggests the importance to shift towards human-relevant and multidisciplinary approaches in these areas of biomedical research. Besides advances in understanding the diseases and using that knowledge for prevention, early detection also requires innovative approaches. The EC has been supporting a large portfolio of projects and initiatives on basic, translational, diagnostic, and clinical research, and has been promoting important monitoring activities to verify how funded research is contributing to scientific innovation and societal impact. These monitoring efforts could also contribute to readdressing funding strategies if needed; allocating future finances in high-quality targeted research for translating newly acquired knowledge into tangible clinical improvements will be key. In this context, more efforts should be made to support data sharing, make well-annotated clinical material available, foster multidisciplinary collaborations, and encourage the dialogue between researchers, funding bodies, government, industry, patients, and the society at large.

## Figures and Tables

**Figure 1 animals-10-01194-f001:**
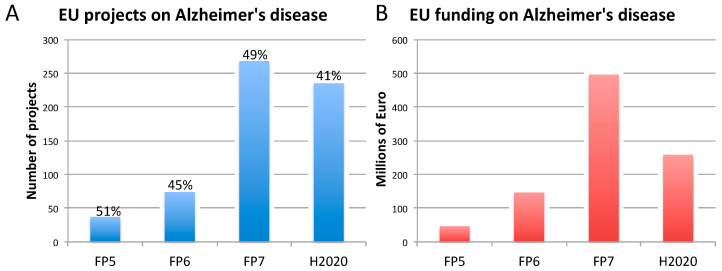
Number of EU-funded projects (**A**) and overall funding (in millions of euro) (**B**) allotted on Alzheimer’s disease-related research across Framework Programme (FP) 5, FP6, FP7 and Horizon 2020 (H2020). The percentages indicate the proportion of projects within a Framework Programme involving also the use of non-human animal models. Data were obtained from CORDIS [128], considering objective, reporting, and results sections, to assess whether non-human animal models were accounted for (H2020 data, as of 10 April 2020). As H2020 was still ongoing at the time of this analysis, H2020 data are not complete.

**Figure 2 animals-10-01194-f002:**
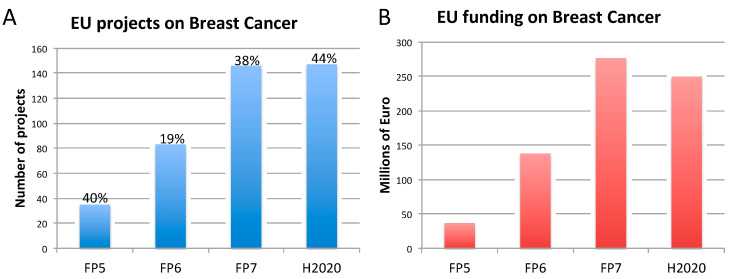
Number of EU-funded projects (**A**) and overall funding (in millions of euro) (**B**) allotted on breast cancer-related research across Framework Programme (FP) 5, FP6, FP7 and Horizon 2020 (H2020). The percentages indicate the proportion of projects within a Framework Programme involving also the use of non-human animal models. Data were obtained from CORDIS [128], considering objective, reporting, and results sections, to assess whether non-human animal models were accounted for. FP7 and H2020 data were obtained using the EC CORDA database (H2020 data, as of May 15, 2019). As H2020 was still ongoing at the time of this analysis, H2020 data are not complete.

**Figure 3 animals-10-01194-f003:**
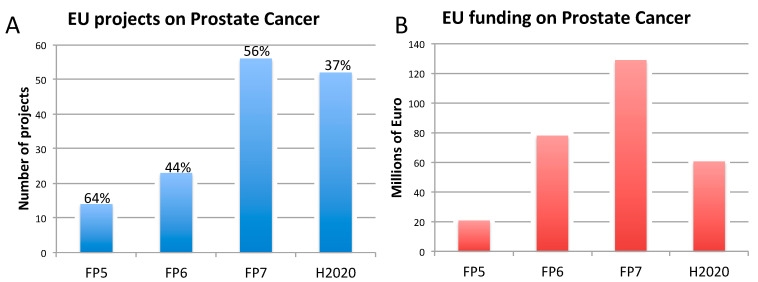
Number of EU-funded projects (**A**) and overall funding (in millions of euro) (**A**) allotted on prostate cancer-related research across Framework Programme (FP) 5, FP6, FP7 and Horizon 2020 (H2020). The percentages indicate the proportion of projects within a Framework Programme involving also the use of non-human animal models. Data were obtained from CORDIS [128], considering objective, reporting, and results sections, to assess whether non-human animal models were accounted for. FP7 and H2020 data were obtained using the EC CORDA database (H2020 data, as of 15 May 2019). As H2020 was still ongoing at the time of this analysis, H2020 data are not complete.

**Table 1 animals-10-01194-t001:** The four most common cancers worldwide (modified from the 2018 world cancer statistics [7]).

Ranking	Cancer Type	New Cases Diagnosed in 2018 (both Sexes)	% of All Cancers (Excluding Non-Melanoma Skin Cancer)
1	Lung	2,093,876	12.3
2	Breast	2,088,849	12.3
3	Colorectal	1,800,977	10.6
4	Prostate	1,276,106	7.5

**Table 2 animals-10-01194-t002:** Proposed list of 18 indicators to measure the outputs and impact of EU-funded biomedical research activities on selected biomedical research area(s).

Category	Indicator	
Funding/Economic	1	Number of projects within a certain EU framework programme (FP)
	2	Value of projects within a certain EU FP
	3	Value of projects from other EU (but non-EC) funding bodies
Dissemination	4	Number of publications (in the frame of a certain research FP) on new scientific insights (e.g., new biomarker, new signaling pathways or mode of action) and whether they were obtained using animal vs. non-animal approaches
	5	Number of publications (in the frame of a certain research FP) on new methods, tools, and approaches (e.g., new diagnostic tool, new treatment approach, new preventive measure) and whether they were obtained using animal vs. non-animal methods and approaches
	6	Number of citations of papers above (i.e., describing either a new scientific insight, or new methods, tools, and approaches)
Scientific and technological	7	Number of patents and whether they were based on animal vs. non-animal findings (e.g., suitable to study selected diseases and/or to test new drugs)
	8	Number of diagnostic tools and whether they were based on animal vs. non-animal findings
	9	Number of approved drugs, treatments, or medical devices and whether they were based on animal vs. non-animal findings
	10	Number of clinical trials for new drugs and whether they were based on animal vs. non-animal findings
	11	Number of new preventive measures and whether they were based on animal vs. non-animal findings
Regulatory and policy	12	Number of public health guidance values/options in regulatory medical-health sectors (e.g., by EMA, national governments, OECD, etc.)
	13	Number of new regulatory policy actions
	14	Number of new non-regulatory targeted policy actions at the national and EU level
Public and social engagement	15	Level of public/social engagement (to disseminate knowledge derived from EU-funded research)
	16	Global indicator(s): Public health trends on selected diseases (e.g., disease prevalence, mortality rate, disease-associated risk factors)
Education, training, and job opportunities	17	New job opportunities resulting from EU-funded research activities
	18	New learning opportunities resulting from EU-funded research activities

EMA: European Medicines Agency; OECD: Organization for Economic Co-operation and Development.

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
