# Peer review of "Alzheimer’s Disease, and Breast and Prostate Cancer Research: Translational Failures and the Importance to Monitor Outputs and Impact of Funded Research"

_animals, 2020, doi:10.3390/ani10071194_

Round 1
Reviewer 1 Report
This is a carefully written and thoughtful piece of work that represents an important addition to our knowledge base and I recommend publication. The authors consider the amount of funding dedicated to different human disease-centred research areas, focusing on those conditions which represent a great societal and economic burden. I found the mixture of European and global data a bit confusing and wonder if it would be possible to focus on Europe- the funding data and the number of projects, and reference to animals used, are all taken from European statistics, but the disease prevalence data and the reference to clinical trials uses data from the NIH, which will include global values/trials. I understand that the authors want to add a global context to their work, but the heavy reliance on European data and the fact that their future activities all seem to be focused on the EU make me think that this better suited as a purely European study.
I would like to see the figures 1-3 discussed more comprehensively in the body of the text. If the authors are trying to make the overall point that the funding spent on projects using animals is essentially wasted money, then the number of projects using animals and the estimated costs of those projects should be a more visible part of the paper - I find that the figures are not enough to bring these points out clearly. These are very important data and in my opinion, they warrant more discussion beyond the figure itself.
The authors mention the use of CORDA in figures 2 and 3, but not figure 1- is this correct?
Lines 152-157, I am not sure what point the authors are making or how this fits the overall argument. I think that this needs to be expanded on, if I understand correctly... I am assuming that the authors making the point that drugs tested in short term animal models, using animals who do not age in the same manner as humans, are not useful for predicting long term effects of treatment on people - but it is not clear to me how this paragraph, as it stands, adds to the overall argument.
Line 161 - it would be helpful to know the extent of this- can you calculate or estimate the actual number or proportion of these drugs (no drug/placebo benefit, terminated on futility analysis/adverse effects?). It seems crucial to have these data- if the authors are arguing for the essential lack of value of animal testing, then it is important to know where adverse effects are responsible for drug termination (as these tend to be a consequence of the preclinical animal testing) compared to other reasons for failure (portfolio rationalisation, for example).
Lines 188-198 - I am not sure what point the authors are making here or how this fits with the overall argument. Is it possible to give any breakdown of funding costs/projects that use either of these mouse types, their use in the examples given already as failed drugs/clinical trials, etc. to try and fit with the aims of the paper.
Line 218-219 - 'only 5% of drug candidates end up advancing through the clinic'- needs citation and confirmation that these data are specific for breast cancer.
Lines 222-225 - it is not clear to me why these are used as examples, or how they add to the main argument presented in the paper. Given it's discovery in rodents, Herceptin could probadly be claimed as a successful product of animal research, so it seems odd to use that example here and I am not sure that I understand the point about these preventing progressive disease - can this be related to the failures of the animal-dependent preclinical research paradigm and short lifespan of animal models?
Lines 226-235 - this is a very important section and I wonder if it needs to be expanded slightly, or at least the link between resistance and a failure of animals to model human disease explored more carefully? As it stands, it is not clear to the reader how drug resistance is related to preclinical testing, are the animal models failing to predict resistance, and so we need to develop 'better' animal models in order to address this- I don't think that this is what the authors are suggesting so this needs clarifying in my opinion.
Lines 236-241 - I am not sure what point the authors are making here and how it fits the overall objective - it seems that the example is not directly relevant to preclinical research using animals. It is not clear what the failures of translation are here - if there are any?
Line 264 - the authors claim that cell passaging 'might' cause clonal section... and this needs a citation and/or stronger language- how often does this happen?
Line 294 - citation needed for the statement about 'replacement with mouse stroma...'.
Line 298 - the point about the futility of testing immunotherapies in immunocompromised animals seems highly important and I think that this should not be buried in the middle of a paragraph and the authors could make more of this. How much funding has gone to these models and how/what have they contributed to therapies?
Line 299 - I don't know why the authors are referring to the co-engraftment of bone marrow to improve model complexity- could they clarify the increased costs here and circle back to the major point about wasted funding?
Lines 303-304 - when the authors make the point about the time required to generate an aggressive model of BC (5 years) could they link that to the animals? Mouse lifespan is around 2 years so this seems to me to represent another good reason for developing models that are not animal-based - to enable these sort of studies.
Lines 308-309 - I am not sure what point the authors are making here - if PDX models are widely used in Pharma, presumably this is not publicly funded, so how does this impact the aims of this study? I think that this statement needs more context. How much EU funding is going to PDX models currently- what is their contribution to (successful) clinical trials and what are the concerns about the wider use of these models in pharma/commercially?
Lines 344-347 - provides a much better explanation of the figure and I would like to see that this is replicated for figures 1 and 2, and also that (even) more detailed explanation is provided. For example, what is the significance of the drop in funding between FP7 and H2020 for PC? And if the animal-dependent projects only decreased 19% for the same programmes- does this represent relatively MORE funding for animal models in H2020? In my opinion, the data presented in the figures are novel and really important and need more attention.
Line 380 - the authors make a point about mouse strains and testosterone levels - but what about in the general human population? Presumably testosterone levels in men are different and so the heterogeneity in mice could be an advantage and offer a raft of animal models to mimic different clinical situations. I think that this statement needs to be explained further - the authors say in line 378-379 that testosterone levels fluctuate between the two species and it may be more relevant to expand on this as evidence for the inability of mice to model a human disease.
Line 393 - the authors state a translation of 5% - do these data refer specially to PC drugs or all oncology applications? It would be good to be specific for PC here if possible.
Line 394- the authors cite reference 111 but I am not sure that this is the most appropriate reference here.
Lines 396-405 - This is a very compelling example but I am not sure I fully understand. The authors claim that the drug was tested in NHP and rats but do not state whether this was safety or efficacy testing - were those animals manipulated to show symptoms of PC or injected with cancer cells prior to treatment? I think that this is an important point to consider - if the animals were used as models for safety (ie toxicity testing) then the clinical trial indicates their failure to predict safety in humans, and likewise, if the animals were disease models, they failed to predict efficacy. Either way, the clinical trial data indicate the futility of this approach. If the authors could estimate costs required to get this one drug to market, that would also be compelling evidence against this animal-dependent paradigm.
Line 424 - I think it is important to clarify what you mean by 'largely funded' - in comparison to the total research funding, or in comparison to other diseases/other countries?
I think that the impacts that the authors outline are clearly presented and appropriate. These do offer an EU-centred vision and that seems to fit the rest of the paper (except in the reporting of NIH clinical trials) but if the paper is to maintain a global perspective, the authors may want to consider, for example, ICCVAM's efforts in this arena.
I think that the discussion is very well written and puts forward many salient points. I would like the authors to clarify what they mean regarding the data that should be shared- data that are submitted with a new drug application would capture preclinical studies but would not include negative studies that fail at the preclinical stage. Might the authors suggest how we could incentivise this?
Lines 529-532 make a very important point, but could bear more weight. Do the authors feel that their research reveals an issue with harm-benefit analysis? Does their calculations regarding costs of research funding and outputs suggest that we need to move away from costly, failing animals in research, or not? I think that this statement would benefit from more context.
Lines 534-552 this section feels a bit to general- these data are good to know but they make limited reference to the specific diseases that form the focus of the study and I think that being more explicit would be useful. Perhaps this could be converted to a table to allow inclusion of all the information presented, plus specifics statistics for AD, PC and BC, where possible.
********* The reviewer 1 sent us these comments also. Please check them (Assistant editor)******************************
Apologies in advance if these comments are what you would expect to go direct to the authors.
General comments: I think that the paper is well written on the whole, and offers an important addition to the field. I am not aware of similar studies that look at the issue with this context, taking account the failures of animal models to translate to humans, and the research presented here could inform and improve future funding strategies.
As indicated to the authors, I think that the paper would benefit from taking an EU-centric approach- mixing global data for disease prevalence with EU funding and animal use does not offer the clearest view. For example, it is confusing to report on clinical trials using the NIH database and it would be helpful to try pull out the EU trials to make this fit with the other parameters discussed.
As outlined more specifically to the authors, some of the paper is informative but, in my opinion is not immediately relevant to the overall aim --eg lines 152-258, 188-198, 222-226. I think it just needs more coherence at times to make sure that the points mentioned are linked to the overall objective of the research. Without being repetitive about the failures of animals models, I think that the authors need to consider how their statements are adding to their arguments and at times (eg lines 226-235, 236-241described above in comments to authors) it is not clear to the reader why these examples are relevant to the main point of the paper.
Specific comments: I have not used or heard of the term 'microfluidics organ on chip' - I think it is more common to use 'microfluidic organs-on-chips' or 'microfluidic organ-on-a-chip'.
I don't think that he subheadings at 2.1, 2.2 and 2.3 are particularly informative and in fact, I don't think that case studies presented calculate the 'rate of translational failure' and suggest that these are rephrased.
I had issues with strange symbols on figures 1-3 on the pdf.
The authors refer to lung cancer in Table 1 but do not go on to discuss this further. However, there is a paper to be submitted to this issue that will specifically consider lung cancer, and I wonder if it would be possible to refer to this (if both papers are accepted, of course!)?
Lines 242-244 do not seem relevant.
The language used in the later part of section 2.2 (eg line 248, line 252) seems to suggest that animal/in vivo models are a required component of preclinical trials and could be used to 'accurately recapitulate clinical features of the disease', and yet my impression was that the authors do not wish for continued funding to go toward animal models that contribute to significant translational failures, or that they fundamentally support the use of animals.
Line 73 - TG animals as reliable proxies of human disease could be referenced.
Line 171- insert 'to be' between 'shown' and 'ineffective'
Lines 274-285 I think that the section on CDX models combines two poor models and this point could be clearer. In fact, I am not really sure what this sections adds to the paper. It seems to me that the disadvantages of the CDX are more closely associated with the cell line used rather than the animal 'host' and I do not understand how this fits the argument of the paper. I wonder if this section is required at all, given that the authors move on to look at PDX models.
Line 438 - change 'rising' to 'rise'
The discussion feels a bit too general at times, the authors list new technologies (eg lines 554-574)but make no reference to how well funded these are currently, or whether there have been any shifts in funding in the framework programmes described. Perhaps it is not possible, but it seems to me that trying to analyse the proportion of research money that has gone toward in vitro studies would be a compelling part of this work.
I agree that monitoring research impacts is crucial - but this alone is unlikely to be sufficient and it might be good for the authors to consider what happens to these metrics? How will they feedback into the funding process/influence the language/direction of funding calls? This would close the circle in my opinion and allows this paper to provide the call for closer monitoring of research funding, models and outputs - to enable value-for-money for EU-funded projects.
Author Response
Reviewer 1:
This is a carefully written and thoughtful piece of work that represents an important addition to our knowledge base and I recommend publication. The authors consider the amount of funding dedicated to different human disease-centred research areas, focusing on those conditions which represent a great societal and economic burden. I found the mixture of European and global data a bit confusing and wonder if it would be possible to focus on Europe- the funding data and the number of projects, and reference to animals used, are all taken from European statistics, but the disease prevalence data and the reference to clinical trials uses data from the NIH, which will include global values/trials. I understand that the authors want to add a global context to their work, but the heavy reliance on European data and the fact that their future activities all seem to be focused on the EU make me think that this better suited as a purely European study.
Response: following reviewer’s comment, we added information about the prevalence of Alzheimer's disease, breast cancer and prostate cancer in Europe, as follows (page 2):
“In 2015, the number of people living with dementia in Europe was estimated to be 10.5 million (with AD accounting for 60 to 80 % of dementia cases) (http://www.aal-europe.eu/is-europe-ready-for-alzheimers/).”
“In Europe (EU-28), in 2016, 97 thousand people died from BC, making up around 7% of all deaths from cancer, and among women, BC accounted for 15.6% of all cancer deaths [8]. With regard to PC, 76.9 thousand men died from PC in the EU-28 in 2016, which corresponds to 5.6% of all deaths from cancer”.
We deleted the information on the clinical trials available from the NIH as possibly misleading; as pointed out by the reviewer, this study is mainly focussed on the European situation, especially with regard to EC-funded research.
I would like to see the figures 1-3 discussed more comprehensively in the body of the text. If the authors are trying to make the overall point that the funding spent on projects using animals is essentially wasted money, then the number of projects using animals and the estimated costs of those projects should be a more visible part of the paper - I find that the figures are not enough to bring these points out clearly. These are very important data and in my opinion, they warrant more discussion beyond the figure itself.
Response: the data presented in Figure 1-3 (now under section 3) and relative text should not lead to the conclusion that that ‘the funding spent on projects using animals is essentially wasted money’. They simply indicate the proportion (percentage) of projects (within each EU framework programme) accounting also (but not exclusively) on studies involving animals. In this revised version, we highlighted this aspect (i.e., not esclusively animal models) in relation to all three diseases (pages 11-12). Therefore, with present data, we cannot draw conclusions about possible direct associations between the proportion of projects considering (also) the use of animal models and translational failure rates globally observed for those diseases. We commented on this relevant aspect in the Discussion (pages 16-17):
“Our analysis of the proportions of EU-funded projects in the field on AD, BC and PC accounting also (but not exclusively) on non-human animal models (Figures 1-3), does not allow drawing conclusions about possible links between methodological approaches selected in EU-funded projects and the translational failure characterizing these three areas of biomedical research. A retrospective assessment of the level of societal impact and innovation derived from those EU-funded projects (accounting and/or not accounting for animal-based studies) by means of indicators, as defined in Table 2, could enable possible correlations about adopted methodological approaches in research and translational outputs. As pointed out in section 3, in-depth interviews with former EU-funded projects participants are currently being carried out by the JRC also to define these possible associations.”.
The authors mention the use of CORDA in figures 2 and 3, but not figure 1- is this correct?
Response: yes, it is correct; the most updated analysis of the H2020 projects and funding on Alzheimer’s disease was based on CORDIS, while H2020 data for breast and prostate cancer were retrieved in the EC database CORDA.
Lines 152-157, I am not sure what point the authors are making or how this fits the overall argument. I think that this needs to be expanded on, if I understand correctly... I am assuming that the authors making the point that drugs tested in short term animal models, using animals who do not age in the same manner as humans, are not useful for predicting long term effects of treatment on people - but it is not clear to me how this paragraph, as it stands, adds to the overall argument.
Response: this paragraph only refers to the limited therapeutic effects of currently available treatments for patients affected by Alzheimer’s disease, which have no impact on the long term prognosis of the disease. As commented by the reviewer, the issue of different life expectancy between Tg mice and humans should be carefully considered when designing preclinical experimentation. Notably, different murine strains show remarkably different lifespan, often premature mortality, along with sex differences in expected lifespan, as summarized by Rae and Brown (Neurosci Biobehav Rev. 2015 Oct;57:238-51.). Although efforts to translate lifespan developmental stages in mice to the equivalent stages for humans have been made considering chronological ages (Flurkey et al., 2007), these comparisons have been based on the C57BL6 mouse, which is one of the murine strains with the longest lifespan (ILAR J. 2011;52(1):4-15.). These differences make direct comparison between preclinical studies using different murine strains quite hard, and the translation of mouse data to human clinical studies questionable. We added this comment (at the end of section 2.1, page 5).
Line 161 - it would be helpful to know the extent of this- can you calculate or estimate the actual number or proportion of these drugs (no drug/placebo benefit, terminated on futility analysis/adverse effects?). It seems crucial to have these data- if the authors are arguing for the essential lack of value of animal testing, then it is important to know where adverse effects are responsible for drug termination (as these tend to be a consequence of the preclinical animal testing) compared to other reasons for failure (portfolio rationalisation, for example).
Response: following reviewer’s comment, we added further details on the 2019 analysis of Alzheimer’s disease drug development pipeline conducted by Cummings et al. by using the clinicaltrials.gov database (page 4):
“In the last 15 years, there have been no approved disease-modifying treatments for AD. In a 2019 analysis of the drug development pipeline for AD (conducted by using the clinicaltrials.gov database), 132 agents were in clinical trials for the treatment of AD (28 agents were in forty-two Phase III trials). Of the several drugs that have completed clinical trial evaluation since the 2018 pipeline analysis conducted by the same group, none of them has shown drug-placebo difference. Those clinical trials have been terminated, often upon futility analysis (which probes the ability of a clinical trial to achieve its objectives [54]), such as for crenezumab, aducanumab, verubecestat, lanabecestat, intranasal insulin, pioglitazone, AZD0530, and ITI-007; others have been terminated due to appearance of adverse effects (e.g., atabecestat). Of the 17 disease-modifying treatments that were in Phase III according to authors’ previous 2018 review, 8 were terminated (as of February 12, 2019) [55].”.
Lines 188-198 - I am not sure what point the authors are making here or how this fits with the overall argument. Is it possible to give any breakdown of funding costs/projects that use either of these mouse types, their use in the examples given already as failed drugs/clinical trials, etc. to try and fit with the aims of the paper.
Response: the aim of this section is merely to comment on the limitations of animal models of AD, being either inbred (such as the senescence accelerate mouse model SAMP8), or genetically modified (Tg) animals. Detailed information about the type(s) (e.g., strains) of animal models used in EC-funded projects analyzed in Figure 1 are not available under the Objective, Reporting and Results sections of the CORDIS database, which was considered to assess whether non-human animal models were accounted in research projects. Therefore (and regrettably), associations between selected murine model types and funding costs/projects cannot be made with currently available data.
Line 218-219 - 'only 5% of drug candidates end up advancing through the clinic'- needs citation and confirmation that these data are specific for breast cancer.
Response: following reviewer’s comment, we reformulated the sentence as follows (beginning on page 6): “However, only 5% of molecules that show anticancer activity in preclinical studies, are approved upon demonstration of sufficient efficacy in Phase III clinical trials [90]. This trend is extremely prevalent especially for VEGF inhibitors (e.g., Bevacizumab), which are used also for BC treatment [91].”.
Lines 222-225 - it is not clear to me why these are used as examples, or how they add to the main argument presented in the paper. Given it's discovery in rodents, Herceptin could probably be claimed as a successful product of animal research, so it seems odd to use that example here and I am not sure that I understand the point about these preventing progressive disease - can this be related to the failures of the animal-dependent preclinical research paradigm and short lifespan of animal models?
Response: as pointed out by the reviewer, despite these treatments have succesfully passed preclinical and clinical phases reaching marketing approval, they have been shown ineffective in some patients in the long term, with development of progressive disease. Assessment of drug efficacy in the long term would not be possible in animal models, considering the way preclinical animal experimentation is generally designed (i.e., using animals, such as mice, with limited life span, and treated for relatively short periods of time), which does not allow predicting possible drug resistance and disease progression.
We commented on these aspects (page 6), as follows:
“This suggests that, despite drugs successfully passing preclinical and clinical phases, reaching marketing approval, they may still prove ineffective in the long term. Assessment of drug efficacy in the long term would not be possible in animal models, considering the way preclinical animal experimentation is generally designed (i.e., based on animals, such as mice, with limited life span, and treated for relatively short periods of time), which does not allow predicting possible drug resistance and disease progression.”
Lines 226-235 - this is a very important section and I wonder if it needs to be expanded slightly, or at least the link between resistance and a failure of animals to model human disease explored more carefully? As it stands, it is not clear to the reader how drug resistance is related to preclinical testing, are the animal models failing to predict resistance, and so we need to develop 'better' animal models in order to address this- I don't think that this is what the authors are suggesting so this needs clarifying in my opinion.
Response: we clarified how iniparib can be considered as a stark example of how incorrect interpretation of preclinical data in animals, along with poor clinical trial design can skew results interpretation and decision making, causing late-stage trial failure. In particular, as highlighted by Mateo et al. (Nat Rev Clin Oncol. 2013 Dec;10(12):688-96.), collected preclinical data did not enable to elucidate the mechanism of action of iniparib before the initiation of clinical trials, and Phase 1 trials did not prove the mechanism of action of this drug. Additionally, inappropriate selection of patients and the lack of implementation and validation of predictive biomarkers, can further contribute to clinical failure. These are just some of the critical factors to carefully consider during the development of anticancer drugs, in order to minimize failures in future late-stage clinical trials. We commented on these aspects (page 6):
“Iniparib is a stark example of how incorrect interpretation of preclinical data in animals, along with poor clinical trial design may skew results interpretation and decision making, leading to late-stage trial failure. As highlighted by Mateo et al. [98], preclinical data could not elucidate on the mechanism of action of iniparib before the initiation of clinical trials, and Phase I trials did not prove the mechanism of action of this drug. Additionally, inappropriate selection of patients and the lack of implementation and validation of predictive biomarkers, can further contribute to clinical failure. These are just some of the critical factors to be carefully considered during the development of anticancer drugs, in order to minimize failures in future late-stage clinical trials.”
Lines 236-241 - I am not sure what point the authors are making here and how it fits the overall objective - it seems that the example is not directly relevant to preclinical research using animals. It is not clear what the failures of translation are here - if there are any?
Response: the very high level of tumor heterogeneity observed in breast cancer (with specific histopathological, genomic and proteomic characteristics, and significant genetic evolution occurring during the metastatic process) is another indication of the possible inadequacy of animal models (in particular, cell line derived xenograft models (CDX) and genetically engineered mouse models (GEMMs)) to reliably mimic all these factors. We added a comment on this aspect after the description of currently available animal models for breast cancer (pages 8-9):
“Considering the very high level of tumor heterogeneity observed in BC, with specific histopathological, genomic and proteomic features, and the genetic evolution frequently observed during metastasis [99], the use of animal models (in particular, CDX and GEMM models) to test new drugs may not be the best methodological approach to account for this complexity and understand BC biology and evolution as it occurs in humans.”.
Line 264 - the authors claim that cell passaging 'might' cause clonal selection... and this needs a citation and/or stronger language- how often does this happen?
Response: it is well known that cell passaging may cause clonal selection of cancer cell lines. Gisselsson et al. (Genes Chromosomes Cancer. 2019 Jul;58(7):452-461.) have comprehensively reported about the possible causes of clonal selection, specifically identifying (i) alterations in telomere function occurring over prolonged in vitro culture, and (ii) population (or genetic) bottlenecks (i.e., a significant decrease in the size of a biologically reproductive population that could be caused by factor(s) limiting the number of cells allowed to proliferate) as frequently neglected phenomena that may cause alterations in cell line genotypes even after few passages in vitro. We added this comment (page 7):
“Gisselsson et al. [105] have comprehensively reported about the possible causes of clonal selection, specifically identifying (i) alterations in telomere function occurring over prolonged in vitro culture, and (ii) population (or genetic) bottlenecks (i.e., a significant decrease in the size of a biologically reproductive population that could be caused by factor(s) limiting the number of cells allowed to proliferate) as frequently neglected phenomena that may cause alterations in (cancer) cell line genotypes even after few passages in vitro.”.
Line 294 - citation needed for the statement about 'replacement with mouse stroma...'.
Response: following reviewer’s comment, we added two references in support to this statement: (DeRose et al. Tumor Grafts Derived From Women With Breast Cancer Authentically Reflect Tumor Pathology, Growth, Metastasis and Disease Outcomes. Nat Med. 2011 Oct 23;17(11):1514-20.) and (Hidalgo et al. Patient-derived Xenograft Models: An Emerging Platform for Translational Cancer Research. Cancer Discov. 2014 Sep;4(9):998-1013.) (page 8).
Line 298 - the point about the futility of testing immunotherapies in immunocompromised animals seems highly important and I think that this should not be buried in the middle of a paragraph and the authors could make more of this. How much funding has gone to these models and how/what have they contributed to therapies?
Response: we further elaborated the issue about the impossibility to test immunotherapies in immunocompromised mice and (in relation to next reviewer’s comment on co-engraftment of human immune cells) the creation of ‘humanized’ mice (page 8), as follows:
“Immunocompromised hosts, such as severely compromised immune-deficient (SCID) mice, non-obese diabetic (NOD)–SCID mice, athymic nude mice, recombination-activating gene 2 (Rag2)-knockout mice, and the NOD/SCID/IL2Rγc−/− mice, are frequently used to generate PDX model of BC [114], as they allow tumor engraftment. However, as these models lack immune system cells, they are not suitable for the preclinical testing of immunotherapies [99]. Murine strains with a human immune system (“humanized mice”), generated by engrafting different types of human leukocytes and purified human hematopoietic stem cells (CD34+) obtained from bone marrow, umbilical cord blood cells, fetal livers or thymus tissues, are currently regarded as suitable models for immunotherapy efficacy testing [114]. However, co-engraftment with human immune cells still presents some limitations (e.g., xenogeneic graft-versus-host responses) [115], and introduces an extra layer of complexity and costs associated with their generation and maintenance [116,117].”.
With regard to cost assessment, it has not been possible to retrieve via CORDIS information about whether funding have (ever) been alloted on studies aimed at testing immunotherapies in immunocompromised animals.
Line 299 - I don't know why the authors are referring to the co-engraftment of bone marrow to improve model complexity- could they clarify the increased costs here and circle back to the major point about wasted funding?
Response: we further elaborated on this aspect (page 8, and also our reply to previous comment):
“However, as these models lack immune system cells, they are not suitable for the preclinical testing of immunotherapies [99]. Murine strains with a human immune system (“humanized mice”), generated by engrafting different types of human leukocytes and purified human hematopoietic stem cells (CD34+) obtained from bone marrow, umbilical cord blood cells, fetal livers or thymus tissues, are currently regarded as suitable models for immunotherapy efficacy testing [114]. However, co-engraftment with human immune cells still presents some limitations (e.g., xenogeneic graft-versus-host responses) [115], and introduces an extra layer of complexity and costs associated with their generation and maintenance [116,117].”
As also commented above, it is not possible to retrieve via CORDIS information about whether/how much funding have been alloted on studies aimed at generating ‘humanized mice’.
Lines 303-304 - when the authors make the point about the time required to generate an aggressive model of BC (5 years) could they link that to the animals? Mouse lifespan is around 2 years so this seems to me to represent another good reason for developing models that are not animal-based - to enable these sort of studies.
Response: as correctly pointed out by the reviewer, one of the main limitations of using mice for (breast) cancer research is their limited lifespan, which does not allow monitoring cancer growth and metastasis. While average mouse life span is about 2 years, in the case of the commonly used NOD SCID murine strain, the average life span is approximately 30 weeks, also due to the frequent development of thymic lymphomas (https://www.jax.org/news-and-insights/jax-blog/2013/july/which-host-strain-should-i-use#). Therefore, these models are not suitable for long-term xenotransplantation studies. We commented on this aspect as follows (page 8):
“The average mouse life span is about 2 years, and, in the case of the commonly used NOD SCID murine strain, life span is approximately 30 weeks, which is often due to the spontaneous development of thymic lymphomas. Therefore, these models are not deemed suitable for long-term xenotransplantation studies [118].”.
Lines 308-309 - I am not sure what point the authors are making here - if PDX models are widely used in Pharma, presumably this is not publicly funded, so how does this impact the aims of this study? I think that this statement needs more context. How much EU funding is going to PDX models currently- what is their contribution to (successful) clinical trials and what are the concerns about the wider use of these models in pharma/commercially?
Response: with currently available information, we are not able to ascertain how much EU funding has been alloted onto studies involving PDX models, their contribution to clinical trials, and the possible concerns associated with their use in pharma industry/commercially. Assessment of these factors would require considerable research and goes beyond the scope of this review/perspective article. As here we refer to EU-funded research, we deleted this sentence to avoid confusion: PDX models are widely used in the pharma industry and many companies are now on the market running pre-clinical studies based on PDX models).
Lines 344-347 - provides a much better explanation of the figure and I would like to see that this is replicated for figures 1 and 2, and also that (even) more detailed explanation is provided. For example, what is the significance of the drop in funding between FP7 and H2020 for PC? And if the animal-dependent projects only decreased 19% for the same programmes- does this represent relatively MORE funding for animal models in H2020? In my opinion, the data presented in the figures are novel and really important and need more attention.
Response: we agree with the reviewers that these are novel and important information. However, these data should not be over-interpreted. In particular, it should be considered that, to date, the H2020 framework programme is still ongoing, and therefore data in Figures 1-3 are not exhaustive, being representative only of the number of projects (and funding) available at the time this analysis was conducted using CORDIS or CORDA. This explains the only apparent decrease of funded projects under H2020. For this reason, possible speculations about decreasing (as for Figure 1 for Alzheimer, or Figure 3 for prostate cancer, by comparing FP7 and H2020) or increasing trends (as observed in Figure 2, for breast cancer, by comparing FP7 and H2020) in the percentage of projects accounting also for animal studies may be inaccurate or even misleading. We commented on this issue in the discussion (page 16), as follows:
“However, it should be considered that to date, the H2020 framework program is still ongoing, and therefore data reported in Figures 1-3 are not exhaustive, being representative only of the number of projects (and funding) available at the time this analysis was conducted. This explains the only apparent decrease of funded projects under H2020. For this reason, possible speculations about decreasing or increasing trends in the percentage of projects accounting also for animal studies may be inaccurate or even misleading.”.
Additionally, in Figures 1-3 legends, we specified that “As H2020 FP was still ongoing at the time of this analysis, H2020 data are not complete”.
Line 380 - the authors make a point about mouse strains and testosterone levels - but what about in the general human population? Presumably testosterone levels in men are different and so the heterogeneity in mice could be an advantage and offer a raft of animal models to mimic different clinical situations. I think that this statement needs to be explained further - the authors say in line 378-379 that testosterone levels fluctuate between the two species and it may be more relevant to expand on this as evidence for the inability of mice to model a human disease.
Response: following reviewer’s suggestion, we added the following text in support to the differences in testosterone levels between mice and men (page 10):
“Testosterone levels also fluctuate between the two species, and significant interspecies differences have been described with regard to serum protein binding affinity for androgens, regulation and function of hepatic steroid metabolizing enzymes, and testosterone biosynthesis and metabolism [140]”.
We also further expanded the description of the limitations of transgenic animal models of prostate cancer (page 10):
“Several GEMMs of PC have been developed, by modulating the expression of specific oncogenes or tumor suppressors, growth factors and their receptors, steroid hormone receptors, or regulators of cell cycle and apoptosis, as summarized in [143,144]. An example is provided by the PB-Cre4xPTENloxp/loxp GEMM model of PC, which is considered suitable to study prostate adenocarcinoma development, tumor progression, and metastasis [144]. Similarly, the Tg adenocarcinoma of the mouse prostate (TRAMP) model has been shown to recapitulate both the pre-neoplastic and metastasic stages of PC [145]. However, these models also present intrinsic limitations [146]. In particular, the TRAMP model develops primarily neuroendocrine tumors, is based on an androgen-dependent promoter, rarely undergoes bone metastasis, and exhibits relatively short kinetics opposite to the typically slow development of PC in humans [147]. Along the same line, the PTEN conditional model described above has been shown to develop senescence, which limits cancer progression, and does not develop metastases [147]. These differences in how PC spontaneously develops and evolves in humans and how it is artificially recreated in animals can possibly explain why drugs that are effective in mice (and other animals) are very often not successful in humans.”.
Line 393 - the authors state a translation of 5% - do these data refer specially to PC drugs or all oncology applications? It would be good to be specific for PC here if possible.
Line 394- the authors cite reference 111 but I am not sure that this is the most appropriate reference here.
Response: we could not find a specific publication reporting about the drug attrition rate specifically for prostate cancer. Therefore, we referred here to the publication by Hutchinson and Kirk (Nat Rev Clin Oncol. 2011 Mar 30;8(4):189-90), also cited in the breast cancer section, commenting on drug attrition rate in cancer research, which applies to cancer in general. We also deleted reference 111 at the end of the sentence. Now the sentence reads as follows (page 10):
“As commented in section 2.2., only about 5% of anticancer therapies tested in animals demonstrate sufficient efficacy in phase III clinical trials and are ultimately approved for clinical use [90].”.
Lines 396-405 - This is a very compelling example but I am not sure I fully understand. The authors claim that the drug was tested in NHP and rats but do not state whether this was safety or efficacy testing - were those animals manipulated to show symptoms of PC or injected with cancer cells prior to treatment? I think that this is an important point to consider - if the animals were used as models for safety (ie toxicity testing) then the clinical trial indicates their failure to predict safety in humans, and likewise, if the animals were disease models, they failed to predict efficacy. Either way, the clinical trial data indicate the futility of this approach. If the authors could estimate costs required to get this one drug to market, that would also be compelling evidence against this animal-dependent paradigm.
Response: the preclinical studies considered in this paragraph referred to uncastrated adult male rats (J Steroid Biochem Mol Biol. 2013 Mar; 134():80-91.) and intact and castrated cynomolgus monkeys (J Steroid Biochem Mol Biol. 2012 Apr; 129(3-5):115-28.) used to test Orteronel effect on serum testosterone levels and the weight of androgen-dependent organs. Therefore, these animals have not been specifically used as disease models for prostate cancer, but provided the rational for testing Orteronel in advanced prostate cancer patients. Following reviewer’s suggestion, we revised the paragraph as follows (end of page 10):
“Preclinical studies in animals provided the rationale for testing Orteronel in PC patients. In particular, Orteronel treatment caused a significant suppression of serum testosterone levels, shrinking several androgen-dependent organs in uncastrated rats [149]. Moreover, administration of Orteronel (twice daily) in intact cynomolgus monkeys induced a reduction of serum dehydroepiandrosterone sulfate and testosterone levels vs vehicle control, and in castrated monkeys, such effects were even greater and persisted throughout the treatment period [150,151].”.
With regard to costs, unfortunately we could not find specific information about the estimated costs suitained for Orteronel testing in different phases of clinical trials.
Line 424 - I think it is important to clarify what you mean by 'largely funded' - in comparison to the total research funding, or in comparison to other diseases/other countries?
Response: it is hard to establish direct comparisons between EC funding contribution to these 3 biomedical research areas and funding alloted by other non-EU countries. Looking at the entire H2020 funding programme (covering 2014-2020), about 0.49% of total H2020 funding has been alloted onto Alzheimer’s disease reseach, nearly 0.46% on breast cancer, and about 0.11% on prostate cancer. However, this information does not add anything critical to the narrative of the manuscript. To avoid possible inferences on comparisons between other diseases and/or non-EU countries, we reformulated the paragraph as follows (page 11, section 3):
“In the last two decades, the EU under FP5, FP6, FP7 and the more recent H2020, has invested significantly in research on AD, BC and PC. The majority of these research activities focused on the understanding of the pathophysiology of the diseases, diagnostics and preclinical studies. Animals were used, usually along with other non-animal approaches, in 19-64 percent of projects.”
To improve the flow, Figures 1-3 and text describing EC-fundind contribution for Alzheimer, breast and prostate cancer has been moved now in section 3.
I think that the impacts that the authors outline are clearly presented and appropriate. These do offer an EU-centred vision and that seems to fit the rest of the paper (except in the reporting of NIH clinical trials) but if the paper is to maintain a global perspective, the authors may want to consider, for example, ICCVAM's efforts in this arena.
Response: as specified in our previous reply to reviewer, we decided to delete the information on the clinical trials available from the NIH as possibly misleading; as pointed out by the reviewer, this study is mainly focussed on the European situation, especially with regard to EC-funded research in these three biomedical research areas.
I think that the discussion is very well written and puts forward many salient points. I would like the authors to clarify what they mean regarding the data that should be shared- data that are submitted with a new drug application would capture preclinical studies but would not include negative studies that fail at the preclinical stage. Might the authors suggest how we could incentivise this?
Response: we agree with the reviewer that this is an important point in the discussion and could benefit from additional considerations. As shown by Arrowsmith, 2011 (Nat Rev Drug Discov. 2011 May; 10(5):328-9.), the highest attrition in drug development is observed in the first clinical proof-of-concept (PoC) study, where lack of efficacy or toxicity are frequently observed. For this reason, making first clinical PoC validation studies publicly available might be key to reduce attrition. More collaborative efforts among companies would help reduce help reduce costs and time. Several initiatives have been taken by large companies in the last decade to promote data sharing and transparency. Among these, the so called ‘Medical Publishing Insights and Practices (MPIP) Initiative’ (http://www.mpip-initiative.org/), a collaborative effort among members of several pharmaceutical industries and the International Society for Medical Publication Professionals (ISMPP) (https://www.ismpp.org/) aims at promoting and increasing integrity, trust and transparency in medical publications and general communication, and expanding access to research results. Along the same line, some peer-reviewed journals, such as the Journal of Pharmaceutical Negative Results (www.pnrjournal.com) and the Journal of Negative Results in Biomedicine (https://jnrbm.biomedcentral.com/) focus on publishing original and novel research articles resulting in negative/null results. Other journals, such as BMJ (https://www.bmj.com/) or Plos One (https://journals.plos.org/plosone/) accepts publishing also negative studies. We added these comments in the discussion (page 16).
Lines 529-532 make a very important point, but could bear more weight. Do the authors feel that their research reveals an issue with harm-benefit analysis? Does their calculations regarding costs of research funding and outputs suggest that we need to move away from costly, failing animals in research, or not? I think that this statement would benefit from more context.
Response: our assessment of EU funding in the three selected biomedical research areas is not meant to extrapolate harm-benefit considerations concerning animal research.
The harm-benefit consideration is clearly a fundamental aspect to account for whenever planning and conducting animal experimentation. Along this line, prioritizing the principle of the 3Rs (Replacement, Refinement and Reduction) and, in particular, updating its strategic application in favour of Replacement strategies relevant to human biology should be encouraged to advance biomedical research, as recently commented (ALTEX. 2019;36(3):343-352.). Notwithstanding, a survey aimed at assessing general attitudes to the 3Rs and animal use in biomedical research, suggest a tendency by scientists to prioritize Refinement over Reduction, and Reduction over Replacement, an ‘upturned hierarchy’ of the 3Rs order as originally proposed by Russell and Burch (page 17).
Lines 534-552 this section feels a bit to general- these data are good to know but they make limited reference to the specific diseases that form the focus of the study and I think that being more explicit would be useful. Perhaps this could be converted to a table to allow inclusion of all the information presented, plus specifics statistics for AD, PC and BC, where possible.
Response: EU statistics on animal uses do not allow gathering detailed information about the number of animals specifically used for Alzheimer’s disease, breast cancer and prostate cancer research, but still provide cumulative evidence that these areas of biomedical research (both basic and applied/translational) i.e., CNS disorders (including dementia and Alzheimer) and oncology (including breast and prostate cancer) globally account for a very significant proportion of the number of animals used (including the number of severe procedures and of transgenic animals) and therefore deserve attention and close monitoring. Following reviewer’s comment, we shortened this section, deleting more general considerations (described in the first part of the paragraph) about the percentages of different animal species and total numbers of animals in basic, translational/applied research. We also added this comment at the end of the paragraph (page 17):
“These numbers provide cumulative evidence that these areas of biomedical research (both basic and applied/translational) i.e., CNS disorders (including dementia and AD) and oncology (including BC and PC) globally account for a very significant proportion of animals uses and therefore would deserve attention and close monitoring.”
Additional comments from Reviewer 1:
Apologies in advance if these comments are what you would expect to go direct to the authors.
General comments: I think that the paper is well written on the whole, and offers an important addition to the field. I am not aware of similar studies that look at the issue with this context, taking account the failures of animal models to translate to humans, and the research presented here could inform and improve future funding strategies.
As indicated to the authors, I think that the paper would benefit from taking an EU-centric approach- mixing global data for disease prevalence with EU funding and animal use does not offer the clearest view. For example, it is confusing to report on clinical trials using the NIH database and it would be helpful to try pull out the EU trials to make this fit with the other parameters discussed.
Response: as indicated in our reply to reviewer’s comment, in order to take a more EU-centric approach, we added information about the prevalence of Alzheimer's disease, breast cancer and prostate cancer in Europe, as follows (page 2):
“In 2015, the number of people living with dementia in Europe was estimated to be 10.5 million (with AD accounting for 60 to 80 % of dementia cases) (http://www.aal-europe.eu/is-europe-ready-for-alzheimers/).”
“In Europe (EU-28), in 2016, 97 thousand people died from BC, making up around 7% of all deaths from cancer, and among women, BC accounted for 15.6% of all cancer deaths [8]. With regard to PC, 76.9 thousand men died from PC in the EU-28 in 2016, which corresponds to 5.6% of all deaths from cancer.”
We deleted the information on the clinical trials available from the NIH as possibly misleading; as pointed out by the reviewer, this study is mainly focussed on the European situation, especially with regard to EC-funded research.
As outlined more specifically to the authors, some of the paper is informative but, in my opinion is not immediately relevant to the overall aim --eg lines 152-258, 188-198, 222-226. I think it just needs more coherence at times to make sure that the points mentioned are linked to the overall objective of the research. Without being repetitive about the failures of animals models, I think that the authors need to consider how their statements are adding to their arguments and at times (eg lines 226-235, 236-241 described above in comments to authors) it is not clear to the reader why these examples are relevant to the main point of the paper.
Response: as specified in our replies to reviewer’s comments, we modified these sections to improve coherence and further some of the issues and considerations discussed in the manuscript.
Here the modifications to the manuscript are summarized again:
- Lines 152-158 (original version): this paragraph only refers to the limited therapeutic effects of currently available treatments for patients affected by Alzheimer’s disease, which have no impact on the long term prognosis of the disease. As commented by the reviewer, the issue of different life expectancy between Tg mice and humans should be carefully considered when designing preclinical experimentation. Notably, different murine strains show remarkably different lifespan, often premature mortality, along with sex differences in expected lifespan, as summarized by Rae and Brown (Neurosci Biobehav Rev. 2015 Oct;57:238-51.). Although efforts to translate lifespan developmental stages in mice to the equivalent stages for humans have been made considering chronological ages (Flurkey et al., 2007), these comparisons have been based on the C57BL6 mouse, which is one of the murine strains with the longest lifespan (ILAR J. 2011;52(1):4-15.). These differences make direct comparison between preclinical studies using different murine strains quite hard, and the translation of mouse data to human clinical studies questionable. We added this comment (at the end of section 2.1, page 5).
- Lines 188-198 (original version): the aim of this section is merely to comment on the limitations of animal models of Alzheimer’s disease, being either inbred (such as the senescence accelerate mouse model SAMP8), or genetically modified (Tg) animals. Detailed information about the type(s) (e.g., strains) of animal models used in EC-funded projects analyzed in Figure 1 are not available under the Objective, Reporting and Results sections of the CORDIS database, which was considered to assess whether non-human animal models were accounted in research projects. Therefore (and regrettably), associations between selected murine model types and funding costs/projects cannot be made with currently available data.
- Lines 222-226 (original version): as pointed out by the reviewer, despite these treatments have succesfully passed preclinical and clinical phases reaching marketing approval, they have been shown ineffective in some patients in the long term, with development of progressive disease. Assessment of drug efficacy in the long term would not be possible in animal models, considering the way preclinical animal experimentation is generally designed (i.e., using animals, such as mice, with limited life span, and treated for relatively short periods of time), which does not allow predicting possible drug resistance and disease progression. We commented on these aspects (page 6), as follows: “This suggests that, despite drugs successfully passing preclinical and clinical phases, reaching marketing approval, they may still prove ineffective in the long term. Assessment of drug efficacy in the long term would not be possible in animal models, considering the way preclinical animal experimentation is generally designed (i.e., based on animals, such as mice, with limited life span, and treated for relatively short periods of time), which does not allow predicting possible drug resistance and disease progression.”
- Lines 226-236 (original version): we clarified how iniparib is as a stark example of how incorrect interpretation of preclinical data in animals, along with poor clinical trial design may skew results interpretation and decision-making, leading to late-stage trial failure. In particular, as highlighted by Mateo et al. (Nat Rev Clin Oncol. 2013 Dec;10(12):688-96.), collected preclinical data could not elucidate on the mechanism of action of iniparib before the initiation of clinical trials, and Phase 1 trials did not prove the mechanism of action of this drug. Additionally, inappropriate selection of patients and the lack of implementation and validation of predictive biomarkers, can further contribute to clinical failure. These are just some of the critical factors to be carefully considered during the development of anticancer drugs, in order to minimize failures in future late-stage clinical trials. We commented on these aspects (page 6).
- Lines 236-241 (original version): the very high level of tumor heterogeneity observed in breast cancer (with specific histopathological, genomic and proteomic characteristics, and significant genetic evolution occurring during the metastatic process) is another indication of the possible inadequacy of animal models (in particular, cell line derived xenograft models (CDX) and genetically engineered mouse models (GEMMs)) to reliably mimic all these factors. We added a comment on this aspect after the description of currently available animal models for breast cancer (pages 8-9): “Considering the very high level of tumor heterogeneity observed in BC, with specific histopathological, genomic and proteomic features, and the genetic evolution frequently observed during metastasis [99], the use of animal models (in particular, CDX and GEMM models) to test new drugs may not be the best methodological approach to account for this complexity and understand BC biology and evolution as it occurs in humans.”.
Specific comments:
I have not used or heard of the term 'microfluidics organ on chip' - I think it is more common to use 'microfluidic organs-on-chips' or 'microfluidic organ-on-a-chip'.
Response: we corrected the typo, specifying 'microfluidic organ-on-a-chip'.
I don't think that he subheadings at 2.1, 2.2 and 2.3 are particularly informative and in fact, I don't think that case studies presented calculate the 'rate of translational failure' and suggest that these are rephrased.
Response: following reviewer’s comment, we changed both heading and subheading titles to better reflect the discussed topics:
- Three biomedical research areas characterized by a high rate of translational failure: Alzheimer’s disease, breast cancer and prostate cancer
2.1. Alzheimer’s disease
2.2. Breast cancer
2.3. Prostate cancer
I had issues with strange symbols on figures 1-3 on the pdf.
Response: thanks for spotting this out, we will make sure that conversion into pdf does not generate strange symbols in the graphs.
The authors refer to lung cancer in Table 1 but do not go on to discuss this further. However, there is a paper to be submitted to this issue that will specifically consider lung cancer, and I wonder if it would be possible to refer to this (if both papers are accepted, of course!)?
Response: if the paper of lung cancer will be published, it would be certainly important to cite it the present manuscript. We flagged this in the introduction section with a comment (page 2).
Lines 242-244 do not seem relevant.
Response: as detailed in our previous response to the reviewer, this part reporting more general considerations about most recently published EU animal statistics has been shortened. Now this entire paragraph is as follows (page 17):
“According to this recent report [37], the basic research areas that accounted for the highest numbers of animal uses were nervous system (22% of basic research uses), immune system (17%) and oncology (14%); the applied/translational research areas accounting for the highest numbers of animal uses were human cancer (27% of applied/translational research uses) and human nervous and mental disorders (14%). In basic research, severe procedures represented 9% of uses for both nervous system and oncology, while in applied/translational research, severe procedures represented 9% of uses for human nervous and mental disorders and 7% for human cancers. Tg animals represented 45% of uses in basic research (63% for oncology, 52% for nervous system), and 25% of uses in translational/applied research (42% for human cancers and 29% for human nervous system and mental disorders). These numbers provide cumulative evidence that these areas of biomedical research (both basic and applied/translational) i.e., CNS disorders (including dementia and AD) and oncology (including BC and PC) globally account for a very significant proportion of animals uses and therefore would deserve attention and close monitoring.”
The language used in the later part of section 2.2 (eg line 248, line 252) seems to suggest that animal/in vivo models are a required component of preclinical trials and could be used to 'accurately recapitulate clinical features of the disease', and yet my impression was that the authors do not wish for continued funding to go toward animal models that contribute to significant translational failures, or that they fundamentally support the use of animals.
Response: as pointed out by the reviewer, we believe that funding should be allocated onto studies prioritizing human-relevant approaches. To avoid any possible misunderstanding in our message, we slightly modified this entire paragraph (page 7):
“These data, in combination with validated in vitro or in silico studies, could provide a powerful tool to explore important genomic trends or individual genes involved, possibly enabling personalised medicine approaches.
The key to success in drug development and clinical treatments is the availability of reliable pre-clinical models that accurately recapitulate the relevant clinical features of the disease, and therefore allow to reliably screen anticancer agents with robust clinical correlation.”.
Line 73 - TG animals as reliable proxies of human disease could be referenced.
Response: we modified the paragraph as follows (page 2):
“Transgenic (Tg) animals (mainly rodents) are generally purported to be reliable proxies of human diseases [9], and ‘humanized’ animals accounting for more than one engineered genetic modification are deemed to be more suitable to study complex, multigene disorders [10].”.
Line 171- insert 'to be' between 'shown' and 'ineffective'
Response: we modified the text as recommended (page 4).
Lines 274-285 I think that the section on CDX models combines two poor models and this point could be clearer. In fact, I am not really sure what this sections adds to the paper. It seems to me that the disadvantages of the CDX are more closely associated with the cell line used rather than the animal 'host' and I do not understand how this fits the argument of the paper. I wonder if this section is required at all, given that the authors move on to look at PDX models.
Response: despite the highlighted limitations with the use of CDX models, these models are still widely applied at the early stage of in vivo study both in academia and industry for their user-friendly technique and high repeatability. We decided to keep this paragraph on CDX, but added the following comment (beginning of page 8):
“Despite these limitations, CDX models are still widely applied at early stages of in vivo studies both in academia and industry for their user-friendly technique and high reproducibility, and several CDX mouse models have been made available by animal models providers (e.g., [109,110]).”.
Line 438 - change 'rising' to 'rise'
Response: we changed as indicated (page 13).
The discussion feels a bit too general at times, the authors list new technologies (eg lines 554-574) but make no reference to how well funded these are currently, or whether there have been any shifts in funding in the framework programmes described. Perhaps it is not possible, but it seems to me that trying to analyse the proportion of research money that has gone toward in vitro studies would be a compelling part of this work.
Response: we agree with the reviewer that this is a relevant aspect to consider in the general narrative of the paper. Over the last two decades, the EC has funded more than 200 projects based on the use of alternative approaches, including in vitro models (e.g., 3D cell culture systems, engineered tissues, organ-on-a-chip models, body-on-a-chip), with a total funding of over EUR 700 million. The average annual budget for this activity gradually increased across the consecutive FPs, from 11 million EUR in FP5, to 32 million EUR in FP6, to 47 million EUR in FP7, and 59 million in H2020 (data retrieved from the CORDA EC database). While these numbers refer to the overall investments in the field of new approach methods (NAMs) across different areas of life science research (not exclusively Alzheimer’s disease, breast or prostate cancer), they are indicative of a progressive increase in the use of NAMs and alternatives to animal approaches. We added these comments in the discussion (page 17).
I agree that monitoring research impacts is crucial - but this alone is unlikely to be sufficient and it might be good for the authors to consider what happens to these metrics? How will they feedback into the funding process/influence the language/direction of funding calls? This would close the circle in my opinion and allows this paper to provide the call for closer monitoring of research funding, models and outputs - to enable value-for-money for EU-funded projects.
Response: following reviewer’s comment, we indicated in the discussion that these metrics may be taken into consideration for the design of the strategic planning of the upcoming Horizon Europe funding scheme, contributing to the creation of future calls for proposals, helping prioritize emerging research topics, and could also feed into the mid-term review of the framework programmes (page 18).

Reviewer 2 Report
The manuscript by Francesca Pistollato et al. discusses possible reasons of failures in translation of animal experiments to clinical applications in three non-communicable diseases Alzheimer’s disease, breast and prostate cancer. The authors put major stress on monitoring of funded research. Despite problem of low translability of animal research is extremely important in biomedical research, there are several major problems with the manuscript.
First of all, I did not find any convincing rationale for choosing these three diseases that are completely different in terms of pathomechanisms, treatment strategies, prognosis etc. The authors state that these three conditions were selected because of their high prevalence and high number of animals used. However, it would be much more logical to discuss animal studies in the field of neurodegenerative diseases or malignancies. It would enable discussing some problems specific to one of these large research fields.
The introduction on Alzheimer’s disease is very general. The results of pre-clinical studies that failed to end up with drug introduction are described in a single sentence. I would expect a detailed description of this studies (eg. in a form of a table) and an in-depth analysis of the reasons of these failures. Also the animal models of Alzheimer’s disease are not properly described – the authors just mention a few of them. The statements on the models lack substantial information – it is not enough to state that “promising results obtained by testing the effects of memantine in these models [65,66], have not been corroborated by clinical trials in patients” (page 5 line 186) – this should be much more specific. Problems with AD animal models should be also discussed in more detail, the authors mention some of the problems but there are several other important issues that are missing. In particular, different views on the pathomechanisms of sporadic Alzheimer’s disease (and consequences for choosing of animal models) are not mentioned in the text.
Similar problems are present in the part on cancer models. For example, the authors mention in few sentences a very complex problem of drug resistance. There are also several very general and definite statements that are not supported with detailed discussion, eg. page 7 line 282:” Spontaneous metastases are not common, therefore CDX models do not represent a good in vivo models for these studies”. The description of animal models is also very general and incomplete, eg. on genetically engineered mouse models of breast cancer. Moreover, there is no data specific to this type of cancer (examples of such models in breast cancer etc.).
Finally, it is not explained why improvement of monitoring of funded research may increase the translability of preclinical research. The authors cite numerous criteria that are very important in terms of social and economical impact of the funded research but does not seem to have any effect of the translation potential. This part should be reorganized to fit the first part of the manuscript and clearly explain the possible impact of the proposed monitoring criteria.
Minor issues:
- Figure 4 and Table 2 present virtually the same data
Author Response
Reviewer 2:
The manuscript by Francesca Pistollato et al. discusses possible reasons of failures in translation of animal experiments to clinical applications in three non-communicable diseases Alzheimer’s disease, breast and prostate cancer. The authors put major stress on monitoring of funded research. Despite problem of low translability of animal research is extremely important in biomedical research, there are several major problems with the manuscript.
First of all, I did not find any convincing rationale for choosing these three diseases that are completely different in terms of pathomechanisms, treatment strategies, prognosis etc. The authors state that these three conditions were selected because of their high prevalence and high number of animals used. However, it would be much more logical to discuss animal studies in the field of neurodegenerative diseases or malignancies. It would enable discussing some problems specific to one of these large research fields.
Response: we agree with the reviewers that neurodegenerative diseases (Alzheimer’s disease) and cancer (breast and prostate) are completely different disorders, characterized by different pathomechanisms, risk factors, treatment strategies, prognosis, etc. The selection of these three diseases is not only linked to their high societal prevalence and the high number of animals used, but also the considerable number of EC-driven research initiatives promoted during the last funding schemes (especially during FP7, H2020) and the high level of translational failure in drug development. We slightly modified the text in the introduction (page 3) to better reflect these selection criteria:
“These diseases were selected as representative case studies for NCDs for several reasons: (i) their high prevalence, (ii) the high number of animals used, as indicated by the most recent EU statistics on the number of animals used for scientific purposes [37], (iii) important research investments, and (iv) the high level of translational failure in drug development.”.
The introduction on Alzheimer’s disease is very general. The results of pre-clinical studies that failed to end up with drug introduction are described in a single sentence. I would expect a detailed description of this studies (eg. in a form of a table) and an in-depth analysis of the reasons of these failures.
Response: following reviewer’s comment, we expanded the description regarding the main reasons underlying drug failure in Alzheimer’s disease. In particular, we added further details on the 2019 analysis from Cummings et al. of Alzheimer’s disease drug development pipeline conducted by using the clinicaltrials.gov database (page 4):
“In the last 15 years, there have been no approved disease-modifying treatments for AD. In a 2019 analysis of the drug development pipeline for AD (conducted by using the clinicaltrials.gov database), 132 agents were in clinical trials for the treatment of AD (28 agents were in forty-two Phase III trials). Of the several drugs that have completed clinical trial evaluation since the 2018 pipeline analysis conducted by the same group, none of them has shown drug-placebo difference and therefore clinical trials have been terminated, often upon futility analysis (which probes the ability of a clinical trial to achieve its objectives [54]), such as for crenezumab, aducanumab, verubecestat, lanabecestat, intranasal insulin, pioglitazone, AZD0530, and ITI-007; others have been terminated due to appearance of adverse effects (e.g., atabecestat). Of the 17 disease-modifying treatments that were in Phase III according to authors’ previous 2018 review, 8 were terminated (as of February 12, 2019) [55].”.
Also the animal models of Alzheimer’s disease are not properly described – the authors just mention a few of them. The statements on the models lack substantial information – it is not enough to state that “promising results obtained by testing the effects of memantine in these models [65,66], have not been corroborated by clinical trials in patients” (page 5 line 186) – this should be much more specific.
Response: following reviewer’s comment, we created Supplementary Table 1, providing (i) name, (ii) type of modifications and (iii) neuropathological features of currently available mouse models of Alzheimer’s disease (totally 180) (retrieved from: https://www.alzforum.org/research-models/alzheimers-disease), referring to it in the main text.
We also expanded the description of memantine and AChE inhibitor effects in preclinical studies (in SAMP8 mice), comparing their effects with clinical studies (page 5):
“Currently available treatments have shown beneficial effects in SAMP8 mice. For instance, memantine was found to improve spatial learning and memory and reduce both hippocampal CA1 NFTs and APP levels when administered to SAMP8 mice; additive effects were observed by combining memantine with environmental enrichment [74]. Along the same line, donepezil was found to improve spatial learning and memory ability, increase cerebral glucose metabolism, and reduce Aβ levels in the cortex of SAMP8 mice. When combined with manual acupuncture, additive beneficial effects were observed [75]. However, despite the promising results obtained in preclinical trials, clinical trials have not proven significant beneficial effects, especially in the long term. In particular, memantine treatment has shown unclear positive effects in patients, slowing the process of cognitive loss at most [76]. Combination therapies are considered more promising than individual treatments in slowing cognitive decline; for example, administration of memantine, in combination with AChE inhibitors (e.g., donepezil or galantamine) was shown to provide some behavioral benefits in patients affected by moderate to severe AD [77,78].”.
Problems with AD animal models should be also discussed in more detail, the authors mention some of the problems but there are several other important issues that are missing. In particular, different views on the pathomechanisms of sporadic Alzheimer’s disease (and consequences for choosing of animal models) are not mentioned in the text.
Response: following reviewer’s comment, we have reorganized the entire section describing animal models of AD (both inbred and Tg), adding additional references:
(page 5): “Tg and double Tg animals develop Aβ plaques and associated brain inflammation, undergoing cognitive and behavioral deficits. These models do not produce NFTs, which can be observed along with cognitive deficits in animals engineered to express the mutated tau protein. The triple 3xTg mice (mean lifespan of 12-18 months) express mutated human APP, PSEN1 and tau protein and can generate both Aβ plaques and NFTs, producing also gliosis, synaptic deficits and memory impairment [67].”
(page 5): “Of the 180 available mouse models of AD (Supplementary Table 1 [68]), very few are representative of LOAD [69]. One of them could be the Senescence Accelerated Mouse Prone 8 (SAMP8), with a mean lifespan of 9.7 months, and characterized by a spontaneous accelerated aging phenotype, which is considered more suitable to study brain ageing and LOAD [70]. SAMP8 mice develop Aβ plaques, NFTs, hyperphosphorylated tau, exhibiting spongiosis, gliosis, forebrain cholinergic impairments, and dendritic spine loss [71].”.
(page 5): “Despite the extensive characterization of the SAMP8 and the SAMP8 murine variants, the genes responsible for the accelerated senescence and the exhibited pathological features are almost unknown. Moreover, Aβ plaque formation and cognitive abnormalities in these mice appear to be significantly different from human AD [79].”.
(page 5): “Notably, different murine strains show remarkably different lifespan, often premature mortality, along with sex differences in expected lifespan, as summarized by Rae and Brown [82]. Although efforts to translate lifespan developmental stages in mice to the equivalent stages for humans have been made considering chronological ages [83], these comparisons have been based on the C57BL6 mouse, which is one of the murine strains with the longest lifespan [84]. These differences make direct comparison between preclinical studies using different murine strains quite hard, and the translation of mouse data to human clinical studies questionable. Altogether, these animal-human discrepancies have contributed making basic science research outcomes poorly applicable to human AD [50].”
Similar problems are present in the part on cancer models. For example, the authors mention in few sentences a very complex problem of drug resistance.
Response: following reviewer’s comment, we added some considerations about drug resistance in relation to breast cancer, but applicable also to other tumors (page 6):
“Understanding de novo or acquired resistance to these drugs is one of the biggest challenges in the identification of new effective therapeutic agents. As commented by Moissenko et al. in their perspective article [95], both intra-cellular (e.g., drug metabolism and efflux, target modulations, lesion restoration) and extra-cellular mechanisms (e.g., crosstalk between tumor cells and environmental factors) may be responsible for drug resistance in BC. Although several mechanisms underlying tumor cell resistance to conventional cytotoxic compounds have been elucidated, more research is warranted to elucidate how multidrug resistance occurs in patients with advanced BC [95].”.
There are also several very general and definite statements that are not supported with detailed discussion, eg. page 7 line 282: “Spontaneous metastases are not common, therefore CDX models do not represent a good in vivo models for these studies”.
Response: following reviewer’s comment, we expanded the description of CDX models to study breast cancer, their use and intrinsic limitations (pages 7-8):
“However, CDX models lack the broad molecular transformation events (intra-tumoral heterogeneity) that occur in human tumors and the organotypic tumor microenvironment; therefore, they cannot recapitulate what is observed in patients with particular respect to drug response, and have been shown to poorly predict clinically effective therapies [99]. CDX models very rarely develop spontaneous metastases, making their use to study BC metastasis questionable [102]. Moreover, the cell lines used to generate CDX are generally obtained from highly aggressive tumors or fluids that have been drained from lung metastasis (e.g., MDA-MB-231 cells), which make these models less suitable to study early events in the evolution of the primary tumor [102]. Furthermore, CDX models seem to be more responsive to antiproliferative agents than primary tumor [107].
Despite these limitations, CDX models are still widely applied at early stages of in vivo studies both in academia and industry for their user-friendly technique and high reproducibility, and several CDX mouse models have been made available by animal models providers (e.g., [109,110]).”.
The description of animal models is also very general and incomplete, e.g. on genetically engineered mouse models of breast cancer. Moreover, there is no data specific to this type of cancer (examples of such models in breast cancer etc.).
Response: following reviewer’s comment, we expanded the description on genetically enginnered animals for BC research (pages 8-9):
“Specific Tg animals have been developed in an effort to emulate more closely the genetics of human BC, accounting for the temporal and spatial activation of specific oncogenes and the deletion of tumor suppressors targeted to the murine mammary gland, such as the Cre/loxP conditional BLG-Cre;Brca1F22-24;p53KO model [120]. Mouse and rat models of BC have been customized using the nuclease-based system CRISPR/Cas9, which can target any gene within an eukaryotic genome, and have been made available by different animal vendors. Comprehensive lists of BC GEMMs along with their characteristics have been described in several review articles [121-125].
Considering the very high level of tumor heterogeneity observed in BC, with specific histopathological, genomic and proteomic features, and the genetic evolution frequently observed during metastasis [99], the use of animal models (in particular, CDX and GEMM models) to test new drugs may not be the best methodological approach to account for this complexity and understand BC biology and evolution as it occurs in humans.”.
Along the same line, we provided additional details about prostate cancer genetically enginnered animal models (page 10):
“Several GEMMs of PC have been developed, by modulating the expression of specific oncogenes or tumor suppressors, growth factors and their receptors, steroid hormone receptors, or regulators of cell cycle and apoptosis, as summarized in [143,144]. An example is provided by the PB-Cre4xPTENloxp/loxp GEMM model of PC, which is considered suitable to study prostate adenocarcinoma development, tumor progression, and metastasis [144]. Similarly, the Tg adenocarcinoma of the mouse prostate (TRAMP) model has been shown to recapitulate both the pre-neoplastic and metastasic stages of PC [145]. However, these models also present intrinsic limitations [146]. In particular, the TRAMP model develops primarily neuroendocrine tumors, is based on an androgen-dependent promoter, rarely undergoes bone metastasis, and exhibits relatively short kinetics opposite to the typically slow development of PC in humans [147]. Along the same line, the PTEN conditional model described above has been shown to develop senescence, which limits cancer progression, and does not develop metastases [147]. These differences in how PC spontaneously develops and evolves in humans and how it is artificially recreated in animals can possibly explain why drugs that are effective in mice (and other animals) are very often not successful in humans.”
Finally, it is not explained why improvement of monitoring of funded research may increase the translability of preclinical research. The authors cite numerous criteria that are very important in terms of social and economical impact of the funded research but does not seem to have any effect of the translation potential. This part should be reorganized to fit the first part of the manuscript and clearly explain the possible impact of the proposed monitoring criteria.
Response: following reviewer’s comment, as also indicated in our reply to reviewer 1, we indicated in the discussion that these metrics may be taken into consideration for the design of the strategic planning of the upcoming Horizon Europe funding scheme, contributing to the creation of future calls for proposals, helping prioritize emerging research topics, and could also feed into the mid-term review of the framework programmes (page 18).
Minor issues:
Figure 4 and Table 2 present virtually the same data
Response: following reviewer’s comment, we deleted Figure 4.

Reviewer 3 Report
General Comments
The authors provide a very good summary of the prevalence and incidence of these three diseases around the world, and therefore, it follows, the need for the most effective research into the causes and for treatments and cures.
They also have supplied a comprehensive review of the situation with regard to the science (current and past), and successes/failures and proposed and actual reasons for them, including GM mouse models, among other things. This is very welcome, and should provide a useful paper that will be of interest to many readers, including scientists involved in research into those diseases, with an interest in the use of animals in science generally, as well as those with an interest in human-specific methods and a paradigm shift towards using them routinely and as a first option, in place of animals.
The manuscript is rather long, but perhaps it needs to be – when authors include some criticism of animal models (with its inherent controversy), all bases need to be covered to prevent unwarranted attacks from detractors. However, if some effort could be made to make the manuscript more concise (and therefore more accessible), that would be helpful.
I know that one of the main thrusts of the paper is translational failures, but it was good to see some attention to other efforts that have a positive impact on these areas of research, such as e.g. screening/early detection of breast cancers. I had commented that the manuscript would be much improved with the inclusion—however brief, even a paragraph—of some examples of where human-specific research methods have already made and are making a difference, and to some degree counteracted the failed translation of animal-based data, but then, later, I saw (pleasingly) that there is a brief discussion of the benefits of using human-specific methods in the Discussion. I think the reader would be well advised, earlier in the manuscript, that this is coming - I was wondering why this hadn’t been done, until I saw it later! Or, alternatively, this section could be moved from the Discussion to earlier in the paper. The section of the manuscript on monitoring innovation and impact is good, by the way.
Regarding the Proposed Indicators for measurement of outputs and impact. Should the authors include a caveat about using citations as a measure? It would. Of course, be wrong to give citations as a simple measure of validity/success when they can be, and often are, negative. For example, a citation can be made to show that an animal approach was a failure. Also, is there anything in the Proposed Indicators to safeguard against claims of success being based on a particular AM or NAM simply because of its use or involvement? Claims are often made for these methods when any success was despite their use, and/or when the ‘success’ was speculative to some degree. Overall, I think some caveats and suggestions for nuance and improvement need to be included.
The Discussion is a good summary of salient points, and introduces some useful recommendations.
I think the Conclusion, given what has gone before in the manuscript, could and should be more explicit in mentioning the poor translation of animal models to clinical benefit.
Specific points:
144-146: A little detail of the “encouraging” results, and failures, would be useful.
460-onwards: This section needs to include some caveats about researcher surveys! What proportion of researchers responded? Perhaps they had a reason to do so or not to do so: to highlight (or to exaggerate) the impact of their research; or to hide the failure/lack of impact of their research? How is this feedback and claimed impact tested/validated? How much was speculative? If speculative, with or without foundation? Significant or poor foundation? This reminds me of the Bateson Report in the UK into nonhuman primate research, of which one criticism (among many) was that the value of NHP research was inferred from questionnaires sent to NHP researchers. It stands to reason that researchers, of any kind, are going to convey their own research as being of great importance and value. If the authors of this manuscript can provide anything to add to it that might make the reader’s perception of this survey more robust, that would add much confidence in it. If there is nothing of the kind to add, then some caveats must be included.
Other Points
The hyperlink for reference #5 didn’t work, even with some ‘tweaking’. With some Googling, I found this to be the new link, I think:
https://www.ons.gov.uk/peoplepopulationandcommunity/birthsdeathsandmarriages/deaths/bulletins/deathsregisteredinenglandandwalesseriesdr/2017
and not, as in the bibliography:
https://http://www.ons.gov.uk/peoplepopulationandcommunity/birthsdeathsandmarriages/deaths/bullet ins/deathsregisteredinenglandandwalesseriesdr/2017 - dementia-and-alzheimer-disease-remained-the-leading-cause-of-death-in-2017
There are some subtle language issues that, though very minor, could improve the readability of the manuscript if addressed. Examples include:
line 24: “largely funded” (see also point about lines 62-63, and 424, below).
54: “the first leading cause of deaths”
62-63 and 424: “largely funded” again – does this mean global research in this area has been mainly funded by the EC, or just that research in this area has been greatly funded by the EC?
69: this sentence needs rewriting. E.g. “aiming to” or “with the aim of”; “goal to/goal of”.
133-135: sentence needs attention.
529: this sentence needs rewriting for accuracy and clarification.
Author Response
Reviewer 3:
The authors provide a very good summary of the prevalence and incidence of these three diseases around the world, and therefore, it follows, the need for the most effective research into the causes and for treatments and cures.
They also have supplied a comprehensive review of the situation with regard to the science (current and past), and successes/failures and proposed and actual reasons for them, including GM mouse models, among other things. This is very welcome, and should provide a useful paper that will be of interest to many readers, including scientists involved in research into those diseases, with an interest in the use of animals in science generally, as well as those with an interest in human-specific methods and a paradigm shift towards using them routinely and as a first option, in place of animals.
The manuscript is rather long, but perhaps it needs to be – when authors include some criticism of animal models (with its inherent controversy), all bases need to be covered to prevent unwarranted attacks from detractors. However, if some effort could be made to make the manuscript more concise (and therefore more accessible), that would be helpful.
Response: we wish to thank the reviewer for the positive and encouraging feedbacks on our manuscript. In order to reply to the comments raised by reviewers 1 and 2 we actually had to slightly extend some sections of the paper. The part in the discussion reporting the animal statistics has been partially reduced, deleting the more general considerations (described in the first part of the paragraph) about the percentages of different animal species and total numbers of animals in basic, translational/applied research. Also, Figure 4 has been deleted as considered redundant with information provided in Table 2 (following reviewer #2’s indication).
I know that one of the main thrusts of the paper is translational failures, but it was good to see some attention to other efforts that have a positive impact on these areas of research, such as e.g. screening/early detection of breast cancers. I had commented that the manuscript would be much improved with the inclusion—however brief, even a paragraph—of some examples of where human-specific research methods have already made and are making a difference, and to some degree counteracted the failed translation of animal-based data, but then, later, I saw (pleasingly) that there is a brief discussion of the benefits of using human-specific methods in the Discussion. I think the reader would be well advised, earlier in the manuscript, that this is coming - I was wondering why this hadn’t been done, until I saw it later! Or, alternatively, this section could be moved from the Discussion to earlier in the paper. The section of the manuscript on monitoring innovation and impact is good, by the way.
Response: following reviewer’s suggestion, we anticipated already at the end of the introduction that we also briefly discuss how human-specific methods could already be used to advance basic and translational/applied research in the field of Alzheimer’s disease, breast and prostate cancer research, and could be used to advance drug discovery and testing (page 3).
Regarding the Proposed Indicators for measurement of outputs and impact. Should the authors include a caveat about using citations as a measure? It would. Of course, be wrong to give citations as a simple measure of validity/success when they can be, and often are, negative. For example, a citation can be made to show that an animal approach was a failure.
Response: following reviewer’s recommendation, we commented on the possible caveats about using bibliometric (dissemination) indicators, and in particular, citation indicators, as follows (pages 14-15):
“With regard to bibliometric indicators (dissemination category), special attention should be paid to the possible caveats at stake, especially considering the possible limitations underlying the citation indicator (indicator 6 in Table 2) and the possible issues with its interpretation and validity [165]. As highlighted by Aksnes and colleagues [166], several factors may undermine the use of citations as measurement of performance, which are generally related to the citation process, such as (i) extensive self-citation rates, (ii) so-called ‘negative’ citations (to criticize, correct, and disclaim other works), and (iii) ‘citation circles’ (i.e., researchers citing one another’s work). While these issues are fundamentally inherent in the use of citations as an indicator, they can also be limited, e.g., by adjusting for self-citations, and crosschecking 'negative' citations.”.
Regarding general issues with the use of indicators, carefully designed indicators should also be considered and applied in a complementary way, as set by international best practices on the design and use of composite indicators (http://www.oecd.org/sdd/42495745.pdf), for example by combining dissemination/bibliometric indicators to economic ones, to provide a more accurate monitoring of outputs and impact of EC-funded biomedical research. We clarified this aspect in the text (page 13):
“Strengths and weaknesses of the proposed indicators will be investigated as recommended by internationally recognised best practices [164]. Overall, different types of indicators used in a complementary way may help provide a more accurate monitoring of EC-funded biomedical research.”
Also, is there anything in the Proposed Indicators to safeguard against claims of success being based on a particular AM or NAM simply because of its use or involvement? Claims are often made for these methods when any success was despite their use, and/or when the ‘success’ was speculative to some degree. Overall, I think some caveats and suggestions for nuance and improvement need to be included.
Response: we agree with the reviewer that it may be complicated to safeguard against claims of success being based on a particular method simply because of its use. Apart from the survey we conducted to monitor impact and innovation of EU-funded research in the three selected biomedical research areas, as we already mentioned in the manuscript, we are currently conducting in-depth interviews with a number of projects participants, aimed at elucidating some of the aspects investigated in the survey, in particular those concerning translatability of research, social impact and lay public engagement. This follow up analysis will help clarify some of the replies obtained in the survey, dig more deeply into scientists’ opinions and perception of translational failure in biomedical research, and elucidate whether claims of innovation and impact success have been based on a particular method (animal or non-animal) simply because of its use or involvement in research activities. A synopsis report providing a more detailed analysis of the responses to both the survey and interviews will be published at the end of the process. We commented on this aspect (page 15).
The Discussion is a good summary of salient points, and introduces some useful recommendations. I think the Conclusion, given what has gone before in the manuscript, could and should be more explicit in mentioning the poor translation of animal models to clinical benefit.
Response: following reviewer’s comment, we added the following comment in the Conclusions section (page 18): “The overall poor clinical translation of preclinical data traditionally generated by using animals along with over simplistic in vitro models, suggests the importance to shift towards human relevant and multidisciplinary approaches in these areas of biomedical research.”.
Specific points:
144-146: A little detail of the “encouraging” results, and failures, would be useful.
Response: following reviewer’s comment, we expanded the discussion about positive results observed in animals and the possible failures behind clinical trials in patients (page 4).
“Concerning drug development, despite very encouraging results obtained in preclinical animal models, showing reduction of NFTs, APP or Aβ plaques, often accompanied with significant improvement of spatial learning and memory, therapeutic approaches based on the ‘amyloid cascade’ hypothesis or designed to target tau proteins have generally failed to provide beneficial effects in AD patients [45,46]. One possible explanation for the failure of clinical trials is the time when drugs are being given, considering that Aβ plaques can form decades before the appearance of first cognitive symptoms [47]. Further complicating the relationship between Aβ plaques and AD, studies on post-mortem brains have reported zero or minimal levels of brain Aβ plaques in about 14%–21% of clinically diagnosed patients [48-50].”.
Additional description about positive/encouraging effects observed in mice and controversial effects in patients are described in page 5 (with regard to SAMP8 mice administered with memantine and AChE inhibitors):
“Currently available treatments have shown beneficial effects in SAMP8 mice. For instance, memantine was found to improve spatial learning and memory and reduce both hippocampal CA1 NFTs and APP levels when administered to SAMP8 mice; additive effects were observed by combining memantine with environmental enrichment [74]. Along the same line, donepezil was found to improve spatial learning and memory ability, increase cerebral glucose metabolism, and reduce Aβ levels in the cortex of SAMP8 mice. When combined with manual acupuncture, additive beneficial effects were observed [75]. However, despite the promising results obtained in preclinical trials, clinical trials have not proven significant beneficial effects, especially in the long term. In particular, memantine treatment has shown unclear positive effects in patients, slowing the process of cognitive loss at most [76]. Combination therapies are considered more promising than individual treatments in slowing cognitive decline; for example, administration of memantine, in combination with AChE inhibitors (e.g., donepezil or galantamine) was shown to provide some behavioral benefits in patients affected by moderate to severe AD [77,78].”.
460-onwards: This section needs to include some caveats about researcher surveys! What proportion of researchers responded? Perhaps they had a reason to do so or not to do so: to highlight (or to exaggerate) the impact of their research; or to hide the failure/lack of impact of their research? How is this feedback and claimed impact tested/validated? How much was speculative? If speculative, with or without foundation? Significant or poor foundation? This reminds me of the Bateson Report in the UK into nonhuman primate research, of which one criticism (among many) was that the value of NHP research was inferred from questionnaires sent to NHP researchers. It stands to reason that researchers, of any kind, are going to convey their own research as being of great importance and value. If the authors of this manuscript can provide anything to add to it that might make the reader’s perception of this survey more robust, that would add much confidence in it. If there is nothing of the kind to add, then some caveats must be included.
Response: we agree with reviewer that surveys present several caveats; in particular, surveyed people may often respond differently than they might if they were unaware of the researcher’s interest in them. As a consequence, provided information is self-reported, and respondents may overstate their research impact. We cited the report by Dr. Mark Kasunic titled “Designing an Effective Survey” (available at https://resources.sei.cmu.edu/asset_files/Handbook/2005_002_001_14435.pdf) in support to these considerations and issues. While considering these caveats, as already commented above, we are currently conducting in-depth interviews with a number of projects participants, aimed at elucidating some of the aspects investigated in the survey, in particular those concerning translatability of research, social impact and lay public engagement. This follow up analysis will help clarify some of the replies obtained in the survey, dig more deeply into scientists’ opinions and perception of translational failure in biomedical research, and elucidate whether claims of innovation and impact success have been based on a particular method (animal or non-animal) simply because of its use or involvement in research activities. A synopsis report providing a more detailed analysis of the responses to both the survey and interviews will be published at the end of the process. We commented on all these aspects in page 15.
Other Points
The hyperlink for reference #5 didn’t work, even with some ‘tweaking’. With some Googling, I found this to be the new link, I think:
https://www.ons.gov.uk/peoplepopulationandcommunity/birthsdeathsandmarriages/deaths/bulletins/deathsregisteredinenglandandwalesseriesdr/2017
and not, as in the bibliography:
https://http://www.ons.gov.uk/peoplepopulationandcommunity/birthsdeathsandmarriages/deaths/bullet ins/deathsregisteredinenglandandwalesseriesdr/2017 - dementia-and-alzheimer-disease-remained-the-leading-cause-of-death-in-2017
Response: we corrected the link as pointed out by the reviewer.
There are some subtle language issues that, though very minor, could improve the readability of the manuscript if addressed. Examples include:
line 24: “largely funded” (see also point about lines 62-63, and 424, below).
Response: we changed ‘largely’ with ‘substantially’ or ‘greatly’.
54: “the first leading cause of deaths”
Response: we corrected the typo.
62-63 and 424: “largely funded” again – does this mean global research in this area has been mainly funded by the EC, or just that research in this area has been greatly funded by the EC?
Response: we mean that research in this area has been greatly funded by the EC. We clarified this in the text, see e.g., page 11: “In the last two decades, the EU under FP5, FP6, FP7 and the more recent H2020, has invested significantly in research on AD, BC and PC. The majority of these research activities focused on the understanding of the pathophysiology of the diseases, diagnostics and preclinical studies. Animals were used, usually along with other non-animal approaches, in 19-64 percent of projects.”.
69: this sentence needs rewriting. E.g. “aiming to” or “with the aim of”; “goal to/goal of”.
Response: we rewrote the sentence as follows (page 2): “Animal models of AD, BC and PC have been and are still largely used in some of those projects aiming to recapitulate human disease features,…”.
133-135: sentence needs attention.
Response: this sentence has been expanded and partially rewritten (page 11): “Animal models of AD have been used in several of those projects (between 41% and 51% of these projects, depending of the FP), generally in combination with other non-animal approaches (in vitro, in silico, clinical data) to explore the molecular and cellular mechanisms underlying AD, identify new druggable targets, and test new compounds.”
529: this sentence needs rewriting for accuracy and clarification.
Response: following reviewer’s comment, we rewrote the sentence, and expanded the discussion about harm-benefit analysis, commenting on the 3Rs concept (page 17):
“The use of animals in biomedical research is not only scientifically questionable; it also poses ethical concerns [184-186], often driven upon ‘harm–benefit’ analysis, i.e., expected scientific benefits and societal impact should outweigh the expected harm to the animals [187]. The harm-benefit consideration is clearly a fundamental aspect to account for whenever planning and conducting animal experimentation. Along this line, prioritizing the principle of the 3Rs (Replacement, Refinement and Reduction) and, in particular, updating its strategic application in favour of Replacement strategies relevant to human biology should be encouraged to advance biomedical research, as recently commented [188]. Notwithstanding, a survey aimed at assessing general attitudes to the 3Rs and animal use in biomedical research, suggest a tendency by scientists to prioritize Refinement over Reduction, and Reduction over Replacement, an ‘upturned hierarchy’ of the 3Rs order as originally proposed by Russell and Burch [189].”.

Round 2
Reviewer 2 Report
The authors significantly improved the manuscript. In particular, they clearly explained their idea of reviewing the three non-communicable diseases. They also clearly explained the idea behind monitoring the indicators to measure the output of the projects. Finally, a large number of missing details are included in the revised version of the manuscript.
Author Response
The authors significantly improved the manuscript. In particular, they clearly explained their idea of reviewing the three non-communicable diseases. They also clearly explained the idea behind monitoring the indicators to measure the output of the projects. Finally, a large number of missing details are included in the revised version of the manuscript.
Response: we wish to thank the reviewer for his/her constructive feedbacks on our manuscript, which have helped us to increase its quality and scientific relevance.